# Analgesic effectiveness of wound infiltration with bupivacaine versus a mixture of bupivacaine and tramadol for postoperative pain management among parturients undergoing elective cesarean section under spinal anesthesia: A randomized controlled trial

**Mesay Milkias** *, **Semagn Mekonnen, Hailemariam Getachew, Hailemariam Mulugeta, Siraj Ahmed, Melkamu Kebede, Belete Destaw, Medhanit Melese, Zemedu Aweke**

Department of Anesthesia, Dilla University College of Medicine and Health Sciences, Dilla, Ethiopia

* milikiasssi@gmail.com

## Abstract

### Background

Post-operative pain is among the major post-cesarean problems, with an incidence ranging from 25.5% to 80%. Despite its simplicity, the effectiveness of wound infiltration with a mixture of bupivacaine and tramadol is still unknown. Therefore, this study aims to compare the analgesic effectiveness of wound infiltration with bupivacaine versus a combination of bupivacaine and tramadol for postoperative pain management among parturients undergoing cesarean section under spinal anesthesia.

### Methodology

A double-blind, parallel, randomized controlled trial was conducted on 60 parturients. Parturients were randomized to take either bupivacaine (B = 30) or a combination of bupivacaine and tramadol (BT = 30). The homogeneity of variance was assessed using Levene's test, and normality was assessed using the Shapiro-Wilk test. A numeric rating scale was used to measure pain severity. The independent t-test and the Mann-Whitney U test were used, respectively, for parametric and non-parametric data. A generalized estimating equation was used to assess repeated measurements.

### Result

In total, 60 parturients were analyzed with no dropouts. The severity of pain at the 6th hour was six times greater in the B group compared to the BT group (OR = 6.289, CI, 2.097–18.858, P = 0.001). The mean tramadol consumption was lower in the BT

**Data availability statement:** All relevant data are within the manuscript and its supporting information files.

**Funding:** Dilla University funded this clinical trial. The funders had no role in study design, data collection and analysis, decision to publish, or preparation of the manuscript.

**Competing interests:** The authors have declared that no competing interests exist.

**Abbreviations:** BMI, body mass index; CI, confidence interval; CS, cesarean section; DBP, diastolic blood pressure; HR, heart rate; LA, local anesthetics; NRS, numeric rating scale; OR, odds ratio; PACU, post anesthesia care unit; SA, spinal anesthesia; SBP, systolic blood pressure; $SPO_2$, peripheral capillary oxygen saturation; WI, wound infiltration.

group (140.00 ± 48.066 mg) than in the B group (175.00 ± 34.114 mg), with a statistically significant mean difference of 10.761 (95% CI, 13.459 to 56.541), $t$ (58) = 3.252, P = 0.002, (d = 0.839). The mean first analgesia request time was higher in the mixture of the BT group (367.33 ± 50.099 min) than in the B group (216.33 ± 68.744 min), with a statistically significant difference of 15.530 (95% CI, −182.087 to −119.913), $t$ (58) = 5.6553, P = 0.001.

## Conclusion

Wound infiltration with a combination of bupivacaine and tramadol is more effective than bupivacaine alone for postoperative analgesia in pregnant patients who underwent cesarean section under spinal anesthesia. This clinical trial study was registered at the Pan African Clinical Trial Registry with a unique trial registration number of PACTR202310525672884 (13/10/2023).

## Introduction

A Cesarean section (CS) is a fetal delivery through a uterine incision and an open abdominal incision (laparotomy) [1]. World Health Organization (WHO) data indicates that over 18 million CS is performed annually worldwide. A CS percentage of between five percent and fifteen percent is recommended by the WHO. A rate below 5% suggests that access to medical technology is limited. The WHO also estimated the prevalence of cesarean sections by area, with thirty-six percent in the US, twenty-three percent in Europe, nine percent in Asia, and four percent in Africa [2–4].

The prevalence of CS in Ethiopia was 21% in Butajira General Hospital in 2019 [5]; 29.7% in Jugal Hospital in 2019 [6]; 47.6% in Dessie [7]; 34.3% in Harar [8]; 30.9% public hospitals in Northern Ethiopia [9]; 24.7% in Durame General Hospital [10]; 28.1% in three Hospitals of Southwest Ethiopia [11]; 27.6% in Atta hospital [12]; 38.5% in a national cross-sectional study [13]; 3.5% in another community-based cross-sectional study [14]. The prevalence of post-CS pain is still significant, ranging from 25.5 to 80%, despite improvements in our understanding of the pathogenesis of pain [15].

According to a study by Hussein et al., 89.8% of pain in patients after a cesarean section is moderate to severe. The duration of surgery, the type of anesthetic used, and the type of analgesics administered were all highly associated with the amount of post-cesarean pain [16]. The percentage of moderate-to-severe pain after a CS is 78.4% to 92%, according to another study done in the US, Europe, and Asia. This study suggests that the discrepancy could be caused by barriers in patients' and physcians attitudes as well as the absence of appropriate pain management services [17].

Postoperative pain remains a principal concern for parturients undergoing cesarean sections [18]. Even with the improvements in Enhanced Recovery After Surgery (ERAS) programs, poorly managed postoperative pain can be distressing since it can hinder timely mother-baby bonding and delay recovery [19]. The risk of depression

and chronic postpartum pain was independently correlated with the severity of acute postpartum pain, but not with the mode of delivery. Compared to women who experienced mild postpartum pain, those who experienced severe acute pain were two times more likely to experience persistent pain and three times more likely to develop postpartum depression [20].

Pain following CS has been managed with various postoperative pain management approaches over the years; however, none of them are without adverse effects [15,21,22]. Wound infiltration is a low-cost, simple procedure with a good safety margin that has been employed in resource-limited areas as a multimodal analgesic strategy to treat postoperative pain. However, various published research studies have revealed inconsistent results in lowering postoperative opioid demand and pain severity [23]. Postoperative analgesia can be effectively attained with the use of longer-duration local anesthetic agents [24].

Tramadol is a 2-(dimethylamino)methyl-1-(3ʳ-methoxyphenyl) compound, and it primarily acts on the central nervous system. Its mild agonistic effects on the opioid receptor, its ability to block the absorption of norepinephrine and serotonin (5-HT) in the descending inhibitory pain pathways, and its ability to stimulate the release of 5-HT are at least three different processes by which it exerts its analgesic effects. Research has revealed that tramadol has certain local anesthetic properties when injected into peripheral nerves; however, further studies are needed to reach a definitive conclusion [25].

Nearly all medications can be found in breast milk to some degree [26]. An infant's exposure to these drugs depends on several factors, including the milk-to-plasma ratio, the amount of milk consumed, and the infant's ability to clear the drug from their system. Certain characteristics of drugs can lead to lower concentrations in breast milk, such as a large volume of distribution, low lipid solubility, and ionization in plasma. These factors help reduce potential harm to the neonate by limiting overall drug exposure. Additionally, drugs that have short elimination half-lives and low bioavailability tend to have a reduced impact and enhanced safety for the infant [27].

The effectiveness of wound infiltration with a mixture of bupivacaine and tramadol in lowering the requirement for postoperative opioids and postoperative pain scores has been the subject of conflicting research [28,29]. Therefore, this study aims to compare the analgesic effectiveness of wound site infiltration with bupivacaine versus a mixture of bupivacaine and tramadol for postoperative pain management among parturients who underwent cesarean delivery under spinal anesthesia.

## Methods

### Study design

We have conducted a parallel, double-blind, superiority randomized controlled trial at Dilla University Referral Hospital that follows the ethical standards stated in the Declaration of Helsinki and the Consolidated Standards of Reporting Trials (CONSORT) reporting guidelines [30](S1 Checklist). The parturients were randomly assigned to either the bupivacaine group or the mixture of bupivacaine and tramadol group. All related data used for this clinical trial were registered at the Pan African Clinical Trial Registry with trial registration number PACTR202310525672884 (13/10/2023). The Research Ethics Committee of Dilla University College of Medicine and Health Sciences approved this clinical trial with a unique protocol number (duirb/029/23–05).

### Definitions of the outcome

**Numeric Rating Scale (NRS):** is a reliable method for measuring pain intensity that entails requesting a patient to rate her severity of pain on an eleven-point scale from 0 to 10, where 0 represents no pain and 10 represents the greatest possible pain.
**Total analgesic consumption:** The mean analgesic consumption of drugs in milligrams (mg) consumed in the first twenty-four hours following the surgery.

**Time to first analgesic request:** is the time, measured in minutes, between the end of surgery and when the patient needs analgesics.

## Study participants

The study was conducted at Dilla University Referral Hospital from May 14, 2023, to August 14, 2023. Parturients between 15 and 49 years of age undergoing spinal anesthesia were included in the study. We excluded parturients using alcohol; those diagnosed with chronic pain; pregnant mothers with systemic diseases such as renal impairment, chronic hepatic disease, respiratory disease, and heart disease; morbidly obese patients (BMI > 40 kg/m²); those receiving spinal anesthesia with adjuvants; operations other than lower uterine segment incision; and parturients taking opioids preoperatively.

## Randomization, blinding, and allocation concealment

The parturients were randomly allocated to receive either bupivacaine or a combination of bupivacaine and tramadol after fulfilling the inclusion criteria. The process of simple randomization involved selecting a level from a closed envelope containing either B or BT, where B represents the bupivacaine group and BT denotes a combination of bupivacaine and tramadol. A 1:1 ratio of a randomly generated number by the computer was employed. The patients and data collectors were blinded, and four equally qualified senior anesthetists in charge, who was not part of the study, prepared the medications. A sealed, sequentially labeled envelope was used to ensure allocation concealment.

## Sample size calculation

Based on a prior Indian study, which demonstrated that wound infiltration with a combination of tramadol and bupivacaine extends the painless period, the sample size was determined by selecting analgesic consumption (N1 = 30, $\mu1$ = 386.17 mg, SD1 = 233.84 mg for a group consisting of bupivacaine and tramadol, and N2 = 30, $\mu2$ = 192.50 mg, SD2 = 134.77 mg for bupivacaine alone) [31]. By utilizing an alpha of 0.05, an effect size of 0.4525, a power of 90% (1-β), and a sample size determined using power analysis with G*Power software (version 3.1.9.7) to be 60, we added 10% for the possible dropout rate to the final sample size, which was 53. The sample size was then increased to 60 in total, with thirty parturients allocated to each group.

## Anesthetic management and surgery

The night before the surgery, a pre-anesthetic evaluation was completed, and written informed consent was obtained after a detailed description of the advantages and disadvantages of each type of anesthesia. Before spinal anesthesia was administered on the morning of the surgery, parturients were given 200 mg IV cimetidine and 0.1 mg/kg IV dexamethasone 30 minutes before surgery. Non-invasive blood pressure, pulse oximetry, and electrocardiography (ECG) monitors were attached and recorded. Before spinal anesthesia, 10 mL/kg of isotonic fluid was preloaded into all parturients. Spinal anesthesia was performed in a sitting position with 0.5% hyperbaric bupivacaine at a dose of 0.07 mg/height, and parturients were placed in a supine position immediately after the injection.

Following spinal anesthesia, the parturients' hemodynamics (SpO2, HR, ECG, and BP) were monitored right away. Every ten minutes, measurements and records of the parturients' hemodynamics (SpO2, HR, ECG, and BP) were made, and the grade of intraoperative shivering was recorded. The moment the infant was delivered, uterotonic medications such as misoprostol (400–600 µg sublingual) or oxytocin (10 IU IM; infusion 0.04–0.125 IU/min) were administered [32]. Blood loss, blood transfusions, intraoperative fluid requirements, and complications such as bradycardia, hypotension, drowsiness, nausea, and vomiting were all documented.

When systolic blood pressure (SBP) falls by greater than 20% from the baseline, it is referred to as hypotension. Intravenous fluid is used to treat this condition; if it persists after fluid administration, a vasopressor (adrenaline 0.01 µg/kg) is used, and atropine (0.02 µg/kg) is administered to treat heart rates that are less than 60 beats per minute. The method

used to grade shivering was the Bedside Shivering Assessment Score (BSAS). Partuirents' received IV tramadol 0.5 mg/kg if they had a shivering grade of three or above [33].

Depending on the partuirents' needs, oxygen was given via a nasal prong at a flow rate of two to three liters per minute both during and after surgery. Following the surgery, the last vital sign was documented, and the wound site was in the lower abdomen and transverse and about 1–2 inches above the pubic hairline, was infiltrated with 0.7 mg/kg of 0.25% bupivacaine diluted to 20 ml for group B. The BT group received a mixture of 0.7 mg/kg of 0.25% bupivacaine and 2 mg/kg of tramadol diluted to 20 milliliters by the four experienced obstetricians [31].

Standard care was provided equally for both groups postoperatively. Rescue analgesics were given based on the severity of pain, and the total analgesics (tramadol and diclofenac) administered during the first 24 hours are documented and analyzed as a secondary outcome. A 5-point rating system was used to assess intraoperative nausea and vomiting, and protocol-compliant care was given to any complications (S1 Text). Numeric rating scale (NRS) was used to assess pain severity for 24 hours, which was at T2, T6, T12, T18, and T24 hours-where T is the time.

## Data collection

Data from parturients and clinical charts of parturients who had cesarean sections under intrathecal anesthesia were gathered for the study using standardized questionnaires. The NRS was used to assess the severity of pain at five different time points: T2, T6, T12, T18, and T24. Among the clinical outcomes evaluated were the incidence of intraoperative nausea and vomiting (IONV), the severity of postoperative pain, the occurrence of shivering, and the time when an analgesic was first required.

## Outcomes

The primary outcome was the severity of pain at the 2nd, 6th, 12th, 18th, and 24th hours. The secondary outcomes were total analgesic consumption, the time of the first analgesic request, incidence of shivering, and incidence of intraoperative nausea and vomiting.

## Statistical analysis

The data were entered into the EpiData program version 4.6 and analyzed with IBM SPSS (version 27). The Shapiro-Wilk test and histogram were used to check the normality of the data. Levene's test was used to determine the homogeneity of variance. Welch's t-test was performed for data that violated the assumptions of Levene's test. Box and whisker plots were used to check for outliers. Independent sample t-tests were used for parametric data, and the Mann-Whitney U test was used for nonparametric data. Chi-square and Fisher's exact tests were used for parametric and nonparametric categorical variables, respectively. P-values less than 0.05 are considered statistically significant. Normally distributed data are shown as mean±SD. Non-normally distributed numerical data are presented using the median and interquartile range, and categorical data are represented using percentages.

The 2nd, 6th, 12th, 18th, and 24th hour pain severity was analyzed using a Generalized Estimating Equation (GEE). A GEE was run to compare the difference in pain severity with a numerical rating scale (NRS) within the study groups (bupivacaine versus a mixture of bupivacaine and tramadol). The difference was expressed with statistical significance, odds ratio, and confidence interval regarding the first hour. We followed the modified intention-to-treat analysis principle; only subjects who were randomized and received all the interventions were included in the final analysis.

## Ethics approval and consent to participation

Ethical approval was obtained from the Institutional Review Committee of Dilla University College of Medicine and Health Sciences. We subsequently secured permission from the hospital authority before commencing data collection. The parturients were informed about the study's purpose, procedures, and potential risks and benefits. Informed written consent

was obtained from the selected parturients to confirm their willingness to participate in the study. To maintain confidentiality, the parturients' names were not included in the written questionnaire. The parturients were assured that their refusal to consent or withdrawal from the study would not affect or jeopardize their access to care.

## Results

### Demographic and preoperative characteristics

In all, 60 parturients completed the follow-up and analysis during the study period, and there were no lost follow-ups or missing data, see Study parturients flow chart (Fig 1). A p-value greater than 0.05 indicates that the groups' perioperative and demographic characteristics are similar for the study subjects. A previous CS scar 13 (21.6) in the bupivacaine group and 15 (25) in the bupivacaine and tramadol group) was the most common reason for parturients receiving CS, followed by fetal macrosomia 7 (11.67) in the bupivacaine group and 5 (8.33)in the bupivacaine and tramadol group), as shown in Table 1. All additional datasets are provided as supporting information (see S1 File).

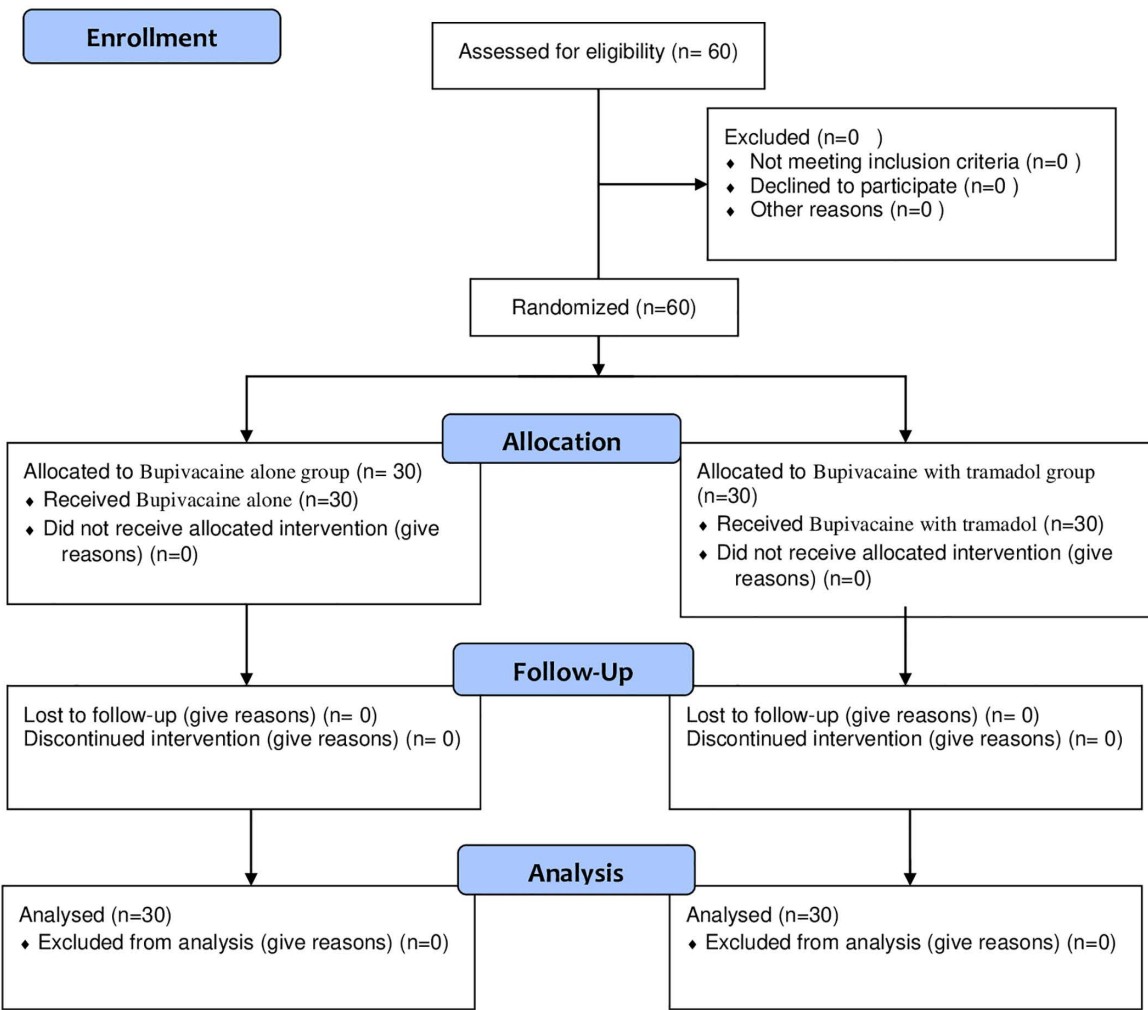

**Fig 1. Study parturients flow chart.**

**Table 1. Demographic and preoperative variables of parturients at Dilla University Referral Hospital, (N = 60).**

| Variable | B(n = 30) | BT(n = 30) | P-value |
|---|---|---|---|
| Age(Yrs) | 26.33 ± 3.36 | 26.47 ± 4.15 | 0.45 |
| Height of the patient(cm) | 159.83 ± 5.3 | 160.97 ± 6.8 | 0.10 |
| Weight of the patient(Kg) | 66.20 ± 8.47 | 69.80 ± 7.53 | 0.40 |
| BMI | 25.87 ± 2.1 | 26.86 ± 2.80 | 0.84 |
| Indication for cesarean section | | | |
| 01 Previous CS scar | 7(11.67) | 9(15) | 0.91 |
| 02 Previous CS scar | 6(10) | 6(10) | |
| Fetal macrosomia | 7(11.67) | 5(8.33) | |
| Mild preeclampsia | 2(3.33) | 2(3.33) | |
| Oligohydramnios | 3(5) | 2(3.33) | |
| Others* | 4(6.67) | 4(6.67) | |
| Baseline vital sign | | | |
| MAP | 87.43 ± 9.98 | 90.07 ± 12.0 | 0.48 |
| HR | 89.97 ± 11.61 | 89.33 ± 1.71 | 0.88 |
| RR | 17.30 ± 1.34 | 17.33 ± 1.62 | 0.21 |
| SPO$_2$ | 96.80 ± 1.32 | 97.07 ± 1.20 | 0.55 |

N (%) = number &percentages, mean (±standard deviation) = mean ± SD. The value is assumed as the mean ± SD for age, weight, and number of patients or frequency for the rest, BMI, Body Mass Index; CS; cesarean section; HR, Heart Rate; MAP; mean arterial pressure, RR, respiratory Rate; SPO$_2$, arterial oxygen saturation; Others*- transverse lie, placenta previa.

## Intraoperative characteristics

The majority of the procedures were done with a sensory level T6 block, with 15 [25] in the group treated with bupivacaine and 17(28.3) in the group treated with bupivacaine and tramadol and a p-value greater than 0.05 indicating that the intra-operative characterstics are similar between the study subject as shown in Table 2.

## Pain severity

The severity of pain in the 6th hour was six times higher in the B group in comparison to the BT group (OR=6.289, CI=2.097–18.858, P=0.001). The severity of pain in the 12th hour was four times higher in the B group in comparison to the BT group (OR=4.313, CI=1.746–10.654, P=0.002). The severity of pain in the 18th hour was 6 times the higher in the B group in comparison to the BT group (OR=6.289, CI=2.328–16.987, P=0.0001).

There was a significant difference across the five time points F (3, 165) =139.8, P<0.001 and there is also a significant difference between groups on pain severity F (1, 58)=268, P<0.001. There was also a significant difference in interaction between time and group F (3, 165) =121, P<0.001(S1 Table). However, there was no significant difference in the 2nd hr and 24th hr follow-up time between groups. Whereas, the mean pain score increased over time between the groups with a significant increment at 18th hour in the control group (B) as compared to the intervention group(BT), see Postoperative pain severity versus follow-up (Fig 2). At the 6th hour postoperatively, the time the uterine exteriorization and pain severity was statistically significant (8.22 ± 1.76 min), (P=0.027).

## Analgesia consumption

The mean tramadol consumption was lower in a mixture of tramadol and bupivacaine group (BT) (140.00 ± 48.06 mg) than bupivacaine group (175.00 ± 34.11 mg) with a statistically significant mean difference(MD) of 10.76 (95% CI, 13.45

Table 2. Intraoperative characteristics of the parturients at Dilla University Referral Hospital, (N=60).

| Intraoperative variables | | B(n=30) | BT(n=30) | P-value |
|---|---|---|---|---|
| Level of sensory block | | | | |
| | T6 | 15(25) | 17(28.3) | 0.39 |
| | T8 | 4(6.67) | 5(8.33) | |
| | Other | 11(18.3) | 8(13.3) | |
| Dose of bupivacaine(ml) | | 12.10±0.89 | 12.18±1.05 | 0.81 |
| Intraoperative shivering | | | | |
| Yes | | 13(21.67) | 8(13.3) | 0.17 |
| No | | 17(28.3) | 22(36.67) | |
| Intraoperative pethidine in mg | | 3.33±10.85 | 4.17±13.26 | 0.56 |
| Duration of surgery in hr. | | 0.6213±0.13 | 0.67±0.15 | 0.17 |
| Duration of anesthesia in hr. | | 0.8897±0.11 | 0.90±0.13 | 0.72 |

B=Bupivacaine; BT=Bupivacaine and Tramadol.

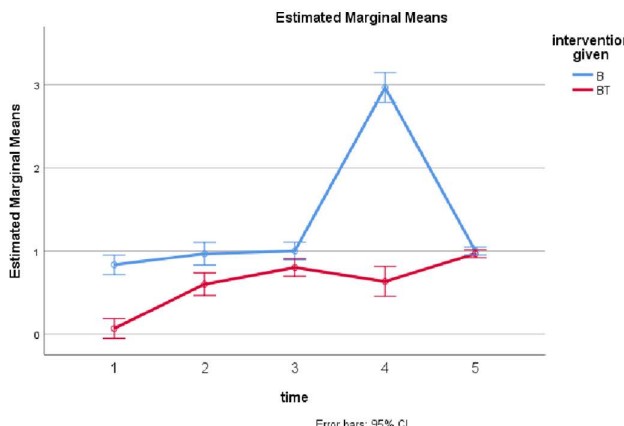

Where= B, Bupivacaine group; BT, Bupivacaine and Tramadol group; the no 1 shows = the pain severity @2hr, the no 2 shows = the pain severity @6hr, the no 3 shows = the pain severity @12hr, the no 4 shows = the pain severity @18hr and the no 5 shows = the pain severity @24hr; the (x-axis) shows follow-up time and the (y-axis) shows the estimated marginal mean of pain severity.

Fig 2. Postoperative pain severity versus follow-up time.

to 56.54), $t$ (58) = 3.25, P=0.002, (d=0.84). There was no significant difference in mean diclofenac consumption between the two groups in a combination of tramadol and bupivacaine group (52.50±48.84 mg) than bupivacaine alone group (57.50±46.95 mg), (P=0.69), (d=0.10).

## First analgesia request time

The mean first analgesia request time in minutes was greater in the mixture of tramadol and bupivacaine group, (Mean±SD) (367.33±50.09 min) than in the bupivacaine group (216.33±68.74 min) with a statistically significant mean difference of 15.53 (95% CI, −182.087 to −119.913), $t$ (58) = 5.6553, P=0.001.

The chi-square test for association was conducted between two groups to compare the incidence of intraoperative shivering, see Incidence of intraoperative shivering (Fig 3). There was no difference between the two groups in terms of the incidence of intraoperative shivering, $X^2(1) = 1.83$, P = 0.17. There was also no statistically significant difference between the two groups in terms of the incidence of intraoperative nausea and vomiting, $X^2(1) = 1.50$, P = 0.50. There was no other complications documented during the study.

## Discussion

In our study, the combination of tramadol and bupivacaine showed a substantial reduction in pain severity compared to bupivacaine alone. Six hours later, the B group had six times more severe pain than the BT group (OR = 6.289, CI = 2.097–18.858, P = 0.001). Compared to the BT group, the B group experienced four times more severe pain at the 12-hour (OR = 4.313, CI = 1.746–10.654, P = 0.002). In the eighteenth hour, the B group experienced six times more severe pain than the BT group (OR = 6.289, CI = 2.328–16.987, P = 0.0001).

Similar to the findings of Sachidananda et al., our study showed a statistically significant difference (p = 0.048) between the mean (±SD) postoperative pain score of the treatment group at 6 and 12 hours after surgery (2.60 ± 0.97) and that of the control group. However, the (mean ±SD) postoperative pain score at 18 and 24 hours was 2.07 ± 0.64 and 2.07 ± 0.37 in the groups receiving bupivacaine and those receiving a combination of bupivacaine and tramadol, respectively, with p = 0.88 [31]. The possible explanation for these discrepancies may be the differences in the study population, pain management modalities, and the preoperative fentanyl used in the study.

According to another study by Gebremedhin et al., which involved 120 patients undergoing lower abdominal surgery, the median (IQR) pain severity (NRS) score was 0.0 (0–0) until 12 hours and 1 (0–5) and 3 (1–5) at 18 and 24 hours in the local wound infiltration (LWI) groups with bupivacaine and tramadol (BT) as opposed to LWI bupivacaine alone (B) in the post-anesthesia care unit and ward. There was a significant difference in pain severity between the BT and B groups at the twelve-, eighteenth-, and twenty-four-hour (p < 0.001) [34]. However, in our study, there was also a significant difference at 2 hours. The discrepancies may be due to differences in the study population, type of surgery, extent of tissue injury, study design, and other confounding variables.

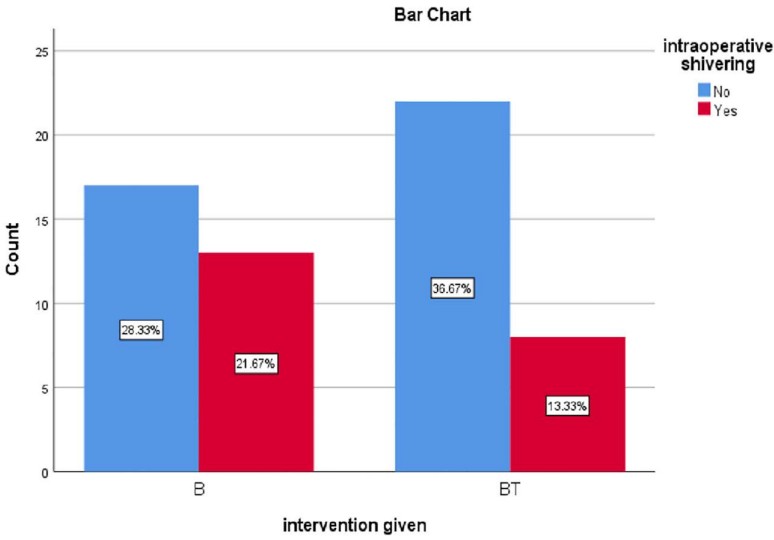

**Fig 3. Incidence of intraoperative shivering.**

Our study found a statistically significant association (P = 0.027) between uterine exteriorization and postoperative pain at 6 hours. At 2 hours, 12 hours, 18 hours, and 24 hours, it was not significant, however. Nafisi's study supports our findings, demonstrating that visceral pain scores were considerably greater in the exteriorization group on the first and second nights compared to the in situ group (66.7 vs. 43.5 on the first night; P < 0.001) and 44.6 vs. 23.9 on the second night (P < 0.001) [35]. Another study by Tan et al. shows that at six hours, exteriorization was linked to a higher incidence of postoperative pain (rated > 5/10) (OR=1.64; 95% CI, 1.31 to 2.03; $I^2$ = 0%). Comparing in situ repair to 24 hours later, there was no difference in pain levels or the requirement for rescue analgesia (OR=2.48; 95% CI, 0.89 to 6.90; $I^2$ = 94%) [36].

A meta-analysis by Zaphiratos et al., in contrast to our results, indicates that there was no statistically significant difference in terms of intraoperative pain (OR = 1.52; 95% CI, 0.86 to 2.71), vomiting (OR = 0.94; 95% CI, 0.66 to 1.35), or nausea (OR = 0.99; 95% CI, 0.74 to 1.34). A more rapid recovery of bowel function was associated with the repair in situ (MD = 3.09 hours; 95% CI, 2.21 to 3.97) [37]. The discrepancies may be due to the small number of studies included in the meta-analysis and the heterogeneity among the studies.

In this study, we found that the mean tramadol consumption in the mixture of bupivacaine and tramadol group was statistically different (140.00 ± 48.066) from the bupivacaine alone group (175.00 ± 34.114), with a statistically significant mean difference of 10.761 (95% CI, 13.459 to 56.541), t (58) = 3.252, (P = 0.002, d = 0.84). The mean diclofenac consumption was not statistically significant between the two groups (BT = 52.50 ± 48.844, B = 57.50 ± 46.955) (P = 0.688, d = 0.104). Additionally, there was no significant difference between the two groups' baseline means for HR, MAP, and SBP.

This finding might be due to tramadol with bupivacaine acting synergistically, which might have provided lower analgesic consumption. The synthetic codeine derivative tramadol is a 4-phenyl piperidine and a receptor agonist. It is serotonergic and 2-adrenergic pathways that mediate the non-opioid process. Norepinephrine and hydroxyl tryptamine are blocked from returning from nerve terminals by it. Several studies published recently claim that local anesthetics and anti-inflammatory drugs have anti-neuronal effects [38].

The efficacy of tramadol for the management of moderate to severe postoperative pain has been revealed in both inpatients and day surgery patients. Most importantly, contrasting with other opioids, tramadol has no clinically significant effects on respiratory or cardiovascular parameters. Tramadol may prove predominantly useful in patients with limited cardiopulmonary function, including the aged, the obese, and smokers; in patients with impaired liver or renal function; and in patients in whom non-steroidal anti-inflammatory drugs are not suggested or need to be used with caution [39]. Additionally, tramadol has a minimal effect on infants who are exposed to less than 3% of the mother's tramadol dose through breast milk, with no evidence of harmful effects. The study suggests that short-term maternal use of tramadol is compatible with breastfeeding, reporting a milk-to-plasma ratio of 2.2 for tramadol and 2.8 for O-desmethyltramadol [40].

The safety of tramadol in neonates is limited due to the small size of the available data. Plasma concentrations of tramadol and its metabolite M1 in neonates, after breastfeeding, are influenced by three main factors: the amount of tramadol and M1 present in breast milk (which depends on maternal drug concentration, maternal metabolism, diffusion, ion trapping, and lipid partitioning), the quantity of breast milk consumed and its bioavailability, and the neonatal clearance of these compounds [41].

It is known that the amount of tramadol excreted into early breast milk is a small percentage of the maternal dose, specifically less than 2.5% per kg of body weight per day. In a study, seventy-five breastfeeding mothers were administered tramadol at a dose of 100 mg every 6 hours following a cesarean birth, in this study exposed infants and the control group of breastfed infants had similar characteristics, such as Apgar scores at birth and scores for neurological and adaptive capacities [42].

The neonate's plasma concentration was 35 times higher than the upper limit found in breastfed infants of mothers prescribed 60 mg of codeine for postpartum pain (0.5–2.2 mcg/L) and 3.5 times the threshold that can cause respiratory depression in neonates (20 mcg/L). A review of the literature on tramadol use during breastfeeding has not identified any

similar adverse events [43]. While both tramadol and its M1 metabolite are present in breast milk, there is no evidence suggesting they carry the same risks associated with the ultra-rapid metabolism of codeine [41]. Several studies have been conducted regarding the safety of opioids in lactating mothers; however, they have small sample sizes, wide confidence intervals, and a high risk of bias [44].

A cohort study conducted by Zewdu et al. on the effectiveness of wound infiltration for patients undergoing elective cesarean section in comparison to the placebo group supports our findings, which indicate that the control group's analgesic use of tramadol and diclofenac over the 24-hour postoperative period was considerably greater than that of the wound site infiltration group (P-value = 0.014) [23]. Consistent with our findings, research by Sarwar, Tasleem, et al. found that the WSI group consumed significantly less tramadol than the placebo group [39]. Additionally, our results are in line with other studies that have reported a much higher request for postoperative analgesia in the bupivacaine group as opposed to the WSI group [24,45].

In our study, the first analgesic request time in minutes for local wound infiltration (LWI) with the BT group is more notable than in the B group. A study conducted by Sachidananda et al. on elective CS patients, comparing wound infiltration with a mixture of tramadol and bupivacaine (BT) versus bupivacaine alone (B), is in line with our results. The mean interval between the first analgesic request following LWI with BT for lower abdominal surgery was 386.17 ± 233.84 min, compared to 192.50 ± 134.7 min in the LWI bupivacaine alone group (P = 0.0002) [31].

Our study's findings are consistent with those of Taksakande et al. In Group BT, the first request for analgesia was made at 390.71 ± 243.74 minutes, while in Group B, it was at 190.05 ± 130.67 minutes. This difference was statistically significant (P = 0.0001). The first rescue analgesia dose was required in Group B earlier than in Group BT, and this difference was also statistically significant (P = 0.0001). When comparing Group B and Group BT, Group BT's postoperative analgesia lasted longer (P = 0.0001) [46].

### Strengths, limitations, and further research

This study was of high quality due to its double-blinded design; however, it has some limitations. These include a lack of control over confounding variables such as the size of the incision, being monocentric, the time of uterine exteriorization, pain only assessed during rest, a shorter duration of postoperative follow-up, tramadol being used for the management of shivering, previous cesarean section scar, the plasma levels of medications not being determined, DNA testing to determine the ultrametabolizers, and different obstetricians performed wound infiltration. We recommend that the researchers conduct further well-designed studies on parturients with coexisting diseases and parturients undergoing emergency surgery.

### Conclusion

This randomized controlled trial reveals that wound infiltration with a mixture of bupivacaine and tramadol (0.7 mg/kg of 0.25% bupivacaine and 2 mg/kg of tramadol) is more effective than bupivacaine alone for postoperative pain management among parturients undergoing cesarean section under spinal anesthesia. Furthermore, the application of a bupivacaine and tramadol mixture to the wound site results in a prolonged period of painlessness and a prolonged duration before the first analgesia request, decreasing the overall consumption of opioids.

### Supporting information

**S1 Checklist. CONSORT 2010 checklist.**
(DOCX)

**S1 File. Data set.**
(XLSX)

**S1 Text. Study protocol in English.**
(PDF)

**S1 Table. Pain severity.**
(DOCX)

## Acknowledgments

We gratefully thank everyone who contributed to this work.

## Author contributions

**Conceptualization:** Mesay Milkias, Semagn Mekonnen, Hailemariam Getachew, Hailemariam Mulugeta, Siraj Ahmed, Melkamu Kebede, Belete Destaw, Medhanit Melese, Zemedu Aweke.

**Data curation:** Mesay Milkias, Semagn Mekonnen, Hailemariam Getachew, Hailemariam Mulugeta, Melkamu Kebede, Medhanit Melese, Zemedu Aweke.

**Formal analysis:** Mesay Milkias, Semagn Mekonnen, Hailemariam Getachew, Hailemariam Mulugeta, Siraj Ahmed, Melkamu Kebede, Belete Destaw, Medhanit Melese, Zemedu Aweke.

**Funding acquisition:** Mesay Milkias, Hailemariam Mulugeta, Siraj Ahmed, Melkamu Kebede, Belete Destaw, Medhanit Melese, Zemedu Aweke.

**Investigation:** Hailemariam Mulugeta, Siraj Ahmed, Melkamu Kebede, Belete Destaw, Medhanit Melese, Zemedu Aweke.

**Methodology:** Mesay Milkias, Semagn Mekonnen, Hailemariam Mulugeta, Belete Destaw, Medhanit Melese, Zemedu Aweke.

**Project administration:** Zemedu Aweke.

**Resources:** Semagn Mekonnen.

**Software:** Belete Destaw, Zemedu Aweke.

**Supervision:** Hailemariam Getachew, Medhanit Melese.

**Visualization:** Mesay Milkias, Hailemariam Getachew, Melkamu Kebede.

**Writing – original draft:** Siraj Ahmed, Melkamu Kebede, Medhanit Melese, Zemedu Aweke.

**Writing – review & editing:** Mesay Milkias, Siraj Ahmed, Zemedu Aweke.

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
