## [Decision Letter · Decision Letter 0]

11 Apr 2025

Dear Dr. Wonte,

Thank you for submitting your manuscript to PLOS ONE. After careful consideration, we feel that it has merit but does not fully meet PLOS ONE’s publication criteria as it currently stands. Therefore, we invite you to submit a revised version of the manuscript that addresses the points raised during the review process.

**ACADEMIC EDITOR:**

Report details on how informed consent for the research was obtained (or explain why consent was not obtained).

NRS and VAS are reported. These outcome measures are not interchangeable. Please explain which was used to measure pain intensity and how this effected the outcome.

Follow the author's instructions on how to submit figures and tables.

We look forward to receiving your revised manuscript.

Kind regards,

Regina (Rianne) L.M. van Boekel

Academic Editor

PLOS ONE

2. We note that you have selected “Clinical Trial” as your article type. PLOS ONE requires that all clinical trials are registered in an appropriate registry (the WHO list of approved registries is at https://www.who.int/clinical-trials-registry-platform/network/primary-registries " https://www.who.int/clinical-trials-registry-platform/network/primary-registries and more information on trial registration is at http://www.icmje.org/about-icmje/faqs/clinical-trials-registration/ ). Please state the name of the registry and the registration number (e.g. ISRCTN or ClinicalTrials.gov ) in the submission data and on the title page of your manuscript. a) Please provide the complete date range for participant recruitment and follow-up in the methods section of your manuscript. b) If you have not yet registered your trial in an appropriate registry, we now require you to do so and will need confirmation of the trial registry number before we can pass your paper to the next stage of review. Please include in the Methods section of your paper your reasons for not registering this study before enrolment of participants started. Please confirm that all related trials are registered by stating: “The authors confirm that all ongoing and related trials for this drug/intervention are registered”. Please see http://journals.plos.org/plosone/s/submission-guidelines#loc-clinical-trials for our policies on clinical trials.

 [Dilla University funded this clinical trial.]. 

5. We note that you have indicated that there are restrictions to data sharing for this study. For studies involving human research participant data or other sensitive data, we encourage authors to share de-identified or anonymized data. However, when data cannot be publicly shared for ethical reasons, we allow authors to make their data sets available upon request. For information on unacceptable data access restrictions, please see http://journals.plos.org/plosone/s/data-availability#loc-unacceptable-data-access-restrictions.

6. In the online submission form, you indicated that [Additional information is provided from the corresponding author upon request.].

7. Please amend either the title on the online submission form (via Edit Submission) or the title in the manuscript so that they are identical.

8. Your ethics statement should only appear in the Methods section of your manuscript. If your ethics statement is written in any section besides the Methods, please delete it from any other section.

Reviewers' comments:

Reviewer's Responses to Questions

**Comments to the Author**

1. Is the manuscript technically sound, and do the data support the conclusions?

Reviewer #1: Yes

Reviewer #2: No

Reviewer #3: Yes

2. Has the statistical analysis been performed appropriately and rigorously?

Reviewer #1: Yes

Reviewer #2: No

Reviewer #3: Yes

3. Have the authors made all data underlying the findings in their manuscript fully available?

Reviewer #1: Yes

Reviewer #2: No

Reviewer #3: Yes

4. Is the manuscript presented in an intelligible fashion and written in standard English?

Reviewer #1: Yes

Reviewer #2: Yes

Reviewer #3: Yes

Reviewer #1: As the statistical reviewer I will focus on methods and reporting. Overall the trial is well described and the methods are appropriate.

1) the power calculation section needs some clarity. clarify the analgesic consumption in the two groups (what unit of measurement) and the power level and alpha at the start - then report the numbers needed to pick up the hypothesised difference (is it 60?). then add the 10% for loss to follow up. I dont' follow how you start with 30+30, then add 10% and end up with 60.

2) the repeated measures approach is appropriate but the section was a bit confusing. it seems like a single repeated measures mixed linear regression model was used? clarify the specifics of the model e.g. clustered errors?

3) There is no information on missing data. were the data complete for all patients in both groups? please state so. if not, would multiple imputation approaches be relevant here?b

Reviewer #2: Mesay Milkias Wonte and colleagues should be congratulated for carrying out a study which aims to alleviate pain in women after Caesarean Delivery.

The authors aim to compare the effect of wound infiltration (WI) with tramadol and bupivacaine versus bupivacaine on its own on pain in women after CSection. They write that previous studies assessed wound infiltration with bupivacaine and tramadol with conflicting results. They cite: (1) Sarwar A TS. Effectiveness of Local Bupivacaine Wound Infiltration in Post Operative Pain Relief After Caesarean Section. J Soc Obstet Gynaecol Pak. 2016;6(3):125-8 and (2) Haliloglu et al . Analgesic efficacy of wound infiltration with tramadol after cesarean delivery under general anesthesia: Randomized trial. J Obstet Gynaecol Res. 2016 Jul;42(7):816-21

I reviewed the abstracts of both papers VERY briefly – they both seem to have found that WI with tramadol effective. I am, therefore, not sure where the results are in conflict with each other. The authors might expand on this point.

This study seems to be a replication of the study carried out by ref 18 : Sachidanandaet at al. Comparison of Analgesic Efficacy of Wound Infiltration with Bupivacaine Versus Mixture of Bupivacaine and Tramadol for Postoperative Pain Relief in Caesarean Section Under Spinal Anaesthesia: A Double-Blind Randomized Trial. Journal of Obstetric Anaesthesia and Critical Care 7(2):p 85-89, Jul–Dec 2017.

It’s a replication in terms of the dose of tramadol that was infiltrated into the wound; size of cohort, method of anesthesia (spinal).

Replicating another study is certainly ethical, but this should be clearly indicated when describing the aims of the study in the Introduction.

Introduction

Third paragraph – I suggest that caring for pain after CSection is complex for the reasons that the authors list but probably a major factor is that mothers, wishing to breastfeed their newborns, and medical and nursing staff are not well acquainted with the medications which are safe vs less safe for lactation.

I suggest that the authors refer to the review by Patricia Lavand’homme Lavand'homme P. Postoperative cesarean pain: real but is it preventable? Curr Opin Anaesthesiol. 2018 Jun;31(3):262-267.

The authors could use information from this review to describe the complexity of caring for women post CSection, including addressing poor recovery and long long outcomes.

Tramadol.

According to Gesseck et al Neonatal Exposure to Tramadol through Mother's Breast Milk. J Anal Toxicol. 2021 Sep 17;45(8):840-846. , use of tramadol during pregnancy is generally avoided and may cause some reversible withdrawal effects in neonates, and its use during lactation is not licensed by the manufacturer.

I assume that if the tramadol is infiltrated to the incision, absorption to the blood stream will be less compared to when the medication is administered systemically. Later in the manuscript, the women seem to have received oral tramadol – and so this might not comply with the current recommendations for lactating women. The authors should address the issue of which opioids are recommended vs not for women after CSection and that there are warnings regarding use of tramadol.

I am aware that different guidelines, internationally, recommend different analgesics for lactating mothers.

I tried to look up international guidelines for recommendations as to which analgesic medications are safe for lactating women. I cite only a few examples here – as to this this thoroughly would require much more work , from the little I have seen, there is little standardization among the guidelines.

LactMed® database – recommend using tramadol for the shortest time possible.

WHO guidelines for lactating mothers BREASTFEEDING AND MATERNAL MEDICATION Recommendations for Drugs in the Eleventh WHO Model List of Essential Drug – the most recent version seems to be from 2002. They do not mention tramadol among the opioids

Roofthooft et al PROSPECT Working Group of the European Society of Regional Anaesthesia and Pain Therapy. PROSPECT guideline for elective caesarean section: an update. Anaesthesia. 2023 Sep;78(9):1170-1171.– recommend wound infiltration but with a local anesthetic alone.

The authors write that tramadol has local anesthetic properties when it is injected into peripheral nerves, citing REFS 13 and 14

I looked up these references. Reference 14 - Subedi et al . An overview of tramadol and its usage in pain management and future perspective. Biomed Pharmacother. 2019 Mar;111:443-451 – states in the Introduction … that use of tramadol is … restricted in pregnant women as well as breast feeding mothers as it may cause birth defects and harm the foetus and to the nursing babies. Tramadol is rated as Category C in the pregnancy risk drug by American Food and Drug Administration. Due to these effects, physicians hould avoid prescribing tramadol to pregnant women and nursing mothers .

This citation does certainly not advocate using tramadol in lactating mothers. Citing it for this statement was probably unintentional.

The second reference seems correct for this statement:

Kaki AM, Al Marakbi W. Post-herniorrhapy infiltration of tramadol versus bupivacaine for postoperative pain relief: a randomized study. Annals of Saudi medicine. 2008;28(3):165-8. 14.

Methods

Why did the authors use both VAS and NRS for assessing pain?

Was pain assessed at rest or movement-related?

Which medication was injected into the spinal catheter?

What do the authors intend to say when they write

….At the 6th hour postoperatively, the time the uterus stayed outside the abdomen and pain severity was statistically significant (8.22±1.767), (P=0.027)

The authors provide summarized results of the pain readings comparing the two groups. They should include a table in which the raw pain scores for each of the assessment time points, for both groups.

Or a graph of the pain scores as a function of time.

Rather than calculate means - which can be misleading - the authors could evaluate the proportion of women who reported scores of eg 3/10 and less at the different time points

I cannot find figure 2.

It will be easier to assess the outcomes related to pain - once the authors address the issues above.

Under Analgesic consumption , page 10 - it seems that patients received systemic tramadol in addition to the infiltration – is this correct? If so - then this seems to contradict recommendations and then it is dubious whether publishing the findings from this study is ethical without extensive discussion of the pros and cons of using tramadol in this population.

Discussion

Page 14 – the authors write

Tramadol may prove predominantly useful in patients with limited cardiopulmonary function, including the aged, the obese, and smokers, in patients with impaired liver or renal function, and in patients in whom non-steroidal anti-inflammatory drugs are not suggested or need to be used with caution (26).

However these patient groups are not included in the current study which focuses on CSection and according to several sources tramadol – at least systemic – is not recommended for women who are breast feeding.

As I wrote above - this is an issue that should be discussed at some length if this manuscript is to be accepted for publication.

Reviewer #3: In the introduction section, I recommend including the prevalence of caesarean sections in Ethiopia, as the study was conducted there and highlights the burden of pain management in caesarean sections and the limited access to opioid analgesics and epidural analgesia in low-resource settings.

Similarly, the authors include, “Research has revealed that tramadol has certain local anesthetic properties when it is injected into peripheral nerves (13, 14).” However, the fact is that wound infiltration, or local infiltration analgesia, involves injecting local anesthetic directly into the tissues surrounding the wound, not the nerves themselves. Therefore, please include references that show the effect of tramadol on wound infiltration if possible. Otherwise, paraphrase the statement if there is a lack of evidence in this regard.

Use a consistent term throughout the documents, either "pregnant" or "parturient." Additionally, use the same abbreviation to describe the two groups (bupivacaine versus a comparison of bupivacaine and tramadol) consistently throughout the documents.

In method sections

Authors mentioned that they used a visual analogue scale and a numeric rating scale to measure pain severity. What is the importance of using two pain assessment scales for a homogenous population?

Redefine total analgesic consumption in the operational definitions as you are using mean tramadol consumption quantity instead of saying analgesic drugs in milligrams.

You mentioned that the anaesthetist in charge, who was not part of the study, prepared the medications. Would you write the details of the person who performed the infiltration procedures? Is it the same obstetrician? Include details of the anatomical locations and techniques where infiltration was performed.

The authors mentioned that patients received IV tramadol at a dose of 0.5 mg/kg if they had a shivering grade of three or above. Does this affect your outcome variables? Intraoperative tramadol may prolong the duration of postoperative analgesia and reduce the mean consumption of postoperative analgesics. At the very least, please acknowledge this point in the limitations section. This might be the reason for the discrepancy with other studies that you have been mentioning in the discussion sections, especially with Sachidananda et al. on the severity of pain scores at different points in time.

Under the Anesthetic Management and Surgery section, you mentioned that a numeric rating scale was used to assess the severity of postoperative pain for 24 hours at T2, T6, T12, T18, and T24 hours, where T represents time. However, in the abstract section, you refer to both a visual analogue scale and a numeric rating scale to measure pain severity, and you also operationalize the definitions of both pain assessment tools. Please clarify which assessment tool you used in the manuscript; probably, you have used NRS.

Please clarify in the outcome section when you evaluated the incidence of shivering during the postoperative period, as the current statement does not specify whether it refers to the intraoperative or postoperative phase.

In the results section, the previous caesarean section scar was an indication for caesarean delivery. Do you think having a CS scar affects your study outcomes, and do you consider it a limitation?

In the intraoperative characteristics, you mentioned that “At the 6th hour postoperatively, the time the uterus stayed outside the abdomen and pain severity was statistically significant (8.22±1.767), (P=0.027).” This information should not be included here. Please either change the heading to "perioperative characteristics" or move this information to another section.

In the discussion section, the authors mentioned that “Similar to the findings of Sachidananda et al., our study showed a statistically significant difference (p = 0.0484) between the mean (±SD) postoperative pain score of the treatment group at 6 and 12 hours after surgery (2.60 ± 0.97) and that of the control group. However, the mean (±SD) postoperative pain score at 18 and 24 hours was 2.07 ± 0.64 and 2.07 ± 0.37 in the groups receiving bupivacaine and those receiving a combination of bupivacaine and tramadol, respectively, with P = 0.8825(18). The possible explanation for these discrepancies may be the differences in the study population.” However, there is no discrepancy in the study populations; both are pregnant women who underwent caesarean delivery and received similar pain management. The only difference is the study location. Please provide other plausible evidence if there is a difference in preoperative analgesics used, doses, and postoperative pain management practices.

Again, the authors noted that “According to another study by Gebremedhin et al., which involved 120 patients undergoing lower abdominal surgery, the median (IQR) pain severity (NRS) score was 0.0 (0-0) until 12 hours, and 1 (0–5) and 3 (1–5) at 18 and 24 hours in the local wound infiltration (LWI) groups with bupivacaine and tramadol (BT) compared to LWI bupivacaine alone (B) in the post-anesthesia care unit and ward. There was a notable difference in pain severity between the BT and B groups at the twelve-, eighteen-, and twenty-four-hour marks (p < 0.001) (44). However, in our study, there was also a significant difference at 2hr. The discrepancies may be due to the difference in study population, study design, and other confounding variables.” In addition to mentioned reasons, include type of surgery as main factor for discrepancy.

I would recommend to paraphrase the limitation as follows:

This study was of high quality due to its double-blinded design; however, it has some limitations. These include a lack of control over confounding variables such as incision size, the study being monocentric, the timing of uterine exteriorization, and a shorter duration of postoperative follow-up. Another limitation is that different obstetricians performed the wound infiltrations, which may have resulted in variations in the procedure of wound infiltration technique.

Please ensure the text is concise and engaging. You don't need to justify every difference; just focus on the pertinent points. Additionally, please proofread the language and grammar throughout the document. Otherwise, well done.

**Do you want your identity to be public for this peer review?** For information about this choice, including consent withdrawal, please see our Privacy Policy

Reviewer #1: No

Reviewer #2: No

Reviewer #3: **Yes: ** Dereje Zewdu Assefa

---

## [Author Response · Author response to Decision Letter 1]

17 May 2025

Rebuttal Letter

Dear Editor,

We would like to thank the editor and reviewers for their careful and thorough reading of this manuscript and for their thoughtful comments and constructive suggestions, which helped us to improve the quality of this manuscript. We have read the research entitled “Analgesic Effectiveness of Wound Infiltration with Bupivacaine versus a Mixture of Bupivacaine and Tramadol for Postoperative Pain Management among Parturients Undergoing Elective Cesarean Section under Spinal Anesthesia”: A Double-Blind Randomized Controlled Trial

We have made all necessary corrections to the comments given by the editor and reviewers one by one as mentioned below. In addition, spelling, grammar, and syntax errors have been corrected.

Editors comments

Comment 1: Please adhere to the guidelines for reporting randomized controlled trials; see author's instructions.

Response: Thanks for the comment. We have followed the author's guidelines to adhere to the guidelines for reporting randomized controlled trials and corrected it.

Comment 2: Report details on how informed consent for the research was obtained (or explain why consent was not obtained).

Response: Thanks for the comment. We have described how the consent was taken in detail as “The parturients were informed about the study's purpose, procedures, and potential risks and benefits. Informed written consent was obtained from the selected parturients to confirm their willingness to participate in the study. To maintain confidentiality, the parturients' names were not included in the written questionnaire. The parturients were assured that their refusal to consent or withdrawal from the study would not affect or jeopardize their access to care.”

Comment 3: NRS and VAS are reported. These outcome measures are not interchangeable. Please explain which was used to measure pain intensity and how this affected the outcome.

Response: Thanks for the comment. It was an editorial problem; we have corrected it. We have assessed pain severity with the numeric rating scale.

Comment 4: Follow the author's instructions on how to submit figures and tables.

Response: Thanks for the comment. We have followed the author's instruction and corrected it. 

Reviewer #1

Comment 1: The power calculation section needs some clarity. Clarify the analgesic consumption in the two groups (what unit of measurement) and the power level and alpha at the start, and then report the numbers needed to pick up the hypothesized difference (is it 60?). then add the 10% for loss to follow up. I dont' follow how you start with 30+30, then add 10% and end up with 60.

Response: Thanks for the comment, We have used G-power to calculate the sample size. The sample size was determined by selecting analgesic consumption (N1 = 30, µ1 = 386.17mg, SD1 = 233.84mg for a group consisting of bupivacaine and tramadol, and N2 = 30, µ2 = 192.50mg, SD2 = 134.77mg for bupivacaine alone). By utilizing alpha 0.05, effect size 0.4525, power 90% (1-β), and sample size determined using power analysis with G power software (version 3.1.9.7) to be 60. We added 10% for the possible dropout rate to the final sample size, which was 53. Then the sample size was increased to 60 in total, and thirty parturients were allocated to each group. Thank you in advance, We have corrected the analgesia consumption between two groups. The mean analgesic consumption of drugs in milligrams (mg) consumed in the first twenty-four hours following the surgery between these two groups to manage pain after assessment.

Comment 2: The repeated measures approach is appropriate, but the section was a bit confusing. It seems like a single repeated measures mixed linear regression model was used? Clarify the specifics of the model, e.g., clustered errors.

Response: Thank you for your comment. We totally agree with you. It was confusing because during the study we considered both VAS as a continuous variable and analyzed it with mixed linear equations and NRS as a categorical variable and analyzed it with general estimating equations to assess whether these two have any difference; however, there was no significant difference between these two pain assessment tools, so we have only used NRS (categorical) for pain assessment and analyzed it with general estimating equations. Now we have addressed all those issues.

Comment 3: There is no information on missing data. Was the data complete for all patients in both groups? Please state so. If not, would multiple imputation approaches be relevant here?

Response: Thank you for comments. We have corrected it; it was an editorial problem. There was no missing data, and the data was complete for both groups; we have mentioned it in the result section (page 7) and figure 1.

Reviewer #2

Comment 1: The authors aim to compare the effect of wound infiltration (WI) with tramadol and bupivacaine versus bupivacaine on its own on pain in women after CSection. They write that previous studies assessed wound infiltration with bupivacaine and tramadol with conflicting results. They cite: (1) Sarwar A TS. Effectiveness of Local Bupivacaine Wound Infiltration in Post Operative Pain Relief After Caesarean Section. J Soc Obstet Gynaecol Pak. 2016;6(3):125-8 and (2) Haliloglu et al . Analgesic efficacy of wound infiltration with tramadol after cesarean delivery under general anesthesia: Randomized trial. J Obstet Gynaecol Res. 2016 Jul;42(7):816-21. I reviewed the abstracts of both papers VERY briefly – they both seem to have found that WI with tramadol effective. I am, therefore, not sure where the results are in conflict with each other. The authors might expand on this point. This study seems to be a replication of the study carried out by ref 18 : Sachidanandaet at al. Comparison of Analgesic Efficacy of Wound Infiltration with Bupivacaine Versus Mixture of Bupivacaine and Tramadol for Postoperative Pain Relief in Caesarean Section Under Spinal Anaesthesia: A Double-Blind Randomized Trial. Journal of Obstetric Anaesthesia and Critical Care 7(2):p 85-89, Jul–Dec 2017. It’s a replication in terms of the dose of tramadol that was infiltrated into the wound; size of cohort, method of anesthesia (spinal). Replicating another study is certainly ethical, but this should be clearly indicated when describing the aims of the study in the introduction.

Response: Thank you for comments, various studies have been conducted in this area, but specifically on cesarean section there are a few studies conducted and no studies was conducted in our country; furthermore those conducted studies have used different analgesics in addition to their intervention, different pain assessment tools, and older analysis method. To determine the effectiveness of intervention, we have calculated the effect size which shows clinical significance, in which none of the previous studies have calculated. We have used Generalized estimating equation (GEE) which has several advantages including flexibility in correlation structure, handling of missing data robustly, focus on population-Averaged effects, and computational effeciecy. None of previously conducted used this analaysis model. Thank you in advance for your observation, we have added the detailed information in introduction section.

Comment 2: Third paragraph – I suggest that caring for pain after C-Section is complex for the reasons that the authors list but probably a major factor is that mothers, wishing to breastfeed their newborns, and medical and nursing staff are not well acquainted with the medications which are safe vs less safe for lactation. I suggest that the authors refer to the review by Patricia Lavand’homme Lavand'homme P. Postoperative cesarean pain: real but is it preventable? Curr Opin Anaesthesiol. 2018 Jun;31(3):262-267. The authors could use information from this review to describe the complexity of caring for women post CSection, including addressing poor recovery and long long outcomes.

Response: Thank you for your suggestion, We have added it, as “Postoperative pain remains a principal concern for parturients undergoing cesarean sections. A cause of suffering, as it can compromise timely mother-baby bonding and delay recovery, despite the advancements of Enhanced Recovery After Surgery (ERAS) programs. The risk of depression and chronic postpartum pain was independently correlated with the severity of acute postpartum pain, but not with the mode of delivery. Compared to women who experienced mild postpartum pain, those who experienced severe acute pain were two times more likely to experience persistent pain and three times more likely to develop postpartum depression”

Comment 3: According to Gesseck et al Neonatal Exposure to Tramadol through Mother's

Breast Milk. JAnal Toxicol. 2021 Sep 17;45(8):840-846. , use of tramadol during pregnancy is

generally avoided and may cause some reversible withdrawal effects in neonates, and its use

during lactation is not licensed by the manufacturer. I assume that if the tramadol is infiltrated to

the incision, absorption to the blood stream will be less compared to when the medication is

administered systemically. Later in the manuscript, the women seem to have received oral

tramadol – and so this might not comply with the current recommendations for lactating women.

The authors should address the issue of which opioids are recommended vs not for women after

C-Section and that there are warnings regarding use of tramadol. I am aware that different

guidelines, internationally, recommend different analgesics for lactating mothers. I tried to look

up international guidelines for recommendations as to which analgesic medications are safe for

lactating women. I cite only a few examples here – as to this this thoroughly would require much

more work , from the little I have seen, there is little standardization among the guidelines.

LactMed® database – recommend using tramadol for the shortest time possible. WHO

guidelines for lactating mothers BREASTFEEDING AND MATERNAL MEDICATION

Recommendations for Drugs in the Eleventh WHO Model List of Essential Drug – the most

recent version seems to be from 2002. They do not mention tramadol among the opioids

Roofthooft et al PROSPECT Working Group of the European Society of Regional Anaesthesia

and Pain Therapy. PROSPECT guideline for elective caesarean section: an update. Anaesthesia.

2023 Sep;78(9):1170-1171.– recommend wound infiltration but with a local anesthetic alone.

The authors write that tramadol has local anesthetic properties when it is injected into peripheral

nerves, citing REFS 13 and 14. I looked up these references. Reference 14 - Subedi et al . An

overview of tramadol and its usage in pain management and future perspective. Biomed

Pharmacother. 2019 Mar;111:443-451 – states in the Introduction … that use of tramadol is …

restricted in pregnant women as well as breast feeding mothers as it may cause birth defects and

harm the foetus and to the nursing babies. Tramadol is rated as Category C in the pregnancy risk

drug by American Food and Drug Administration. Due to these effects, physicians hould avoid

prescribing tramadol to pregnant women and nursing mothers. This citation does certainly not

advocate using tramadol in lactating mothers. Citing it for this statement was probably

unintentional. The second reference seems correct for this statement: Kaki AM, Al Marakbi W.

Post-herniorrhapy infiltration of tramadol versus bupivacaine for postoperative pain relief: a

randomized study. Annals of Saudi medicine. 2008;28(3):165-8. 14.

Response: Thank you for your comment; We totally agree with your concern, it was also our

concern. Here we have administred tramadol subcutaneously which have very less effect on

mothers due its minimal absorption to the blood stream. According to literatures, if the tramadol

is infiltrated to the incision site subcutaneously, absorption to the blood stream will be less

compared to when the medication is administered systemically.All patients allocated to

intervention group(Bupivacaine and Tramadol) received 0.7 mg/kg of 0.25% bupivacaine and 2

mg/kg of tramadol. Neither control nor intervention group received oral tramadol. Thanks in

advance we have corrected reference 14, it was unintentional.

Comment 3: Why did the authors use both VAS and NRS for assessing pain? Was pain assessed

at rest or movement-related? Which medication was injected into the spinal catheter? What do

the authors intend to say when they write? At the 6th hour postoperatively, the time the uterus

stayed outside the abdomen and pain severity was statistically significant (8.22±1.767),(P=0.027)

Response: Thank you for your comment; We have corrected it, only NRS was used to assess

postoperative pain. The postoperative pain was assessed during rest time.

Spinal anesthesia was performed in a sitting position with 0.5% hyperbaric bupivacaine with a

dose of 0.07 mg/height.

We have assessed for the association between uterine exteriorization and pain severity and there

are evidences that reveals uterine exteriorization have effect on post-operative pain severity. The

studies postulates that, this pain is primarily visceral pain, mediated by C-fibers that transmit

pain signals from the uterus to the brain. Uterine exteriorization also affects the vagus nerve,

which innervates the uterine wall, potentially inducing nausea and vomiting. In our study there

was statistically significant association(8.22±1.767), (P=0.027) between post-operative pain

and uterine exteriorization at 6th hour between two groups.

Comment 4: The authors provide summarized results of the pain readings comparing the two

groups. They should include a table in which the raw pain scores for each of the assessment time

points, for both groups. Or a graph of the pain scores as a function of time. Rather than calculate

means - which can be misleading - the authors could evaluate the proportion of women who

reported scores of eg 3/10 and less at the different time points. I cannot find figure 2.

It will be easier to assess the outcomes related to pain - once the authors address the issues

above. Under Analgesic consumption , page 10 - it seems that patients received systemic

tramadol in addition to the infiltration – is this correct? If so - then this seems to contradict

recommendations and then it is dubious whether publishing the findings from this study is ethical

without extensive discussion of the pros and cons of using tramadol in this population.

Response: Thank you for you comment, we have used generalized estimating equations (GEE),

which is advanced analysis model,that mainly focus on population-Averaged effects, rather than

considering single point pain score. This is one of GEEs major advantage over ANOVA.

We have provided figures which are color coded for easy identification from the analysis model,

described details at footnotes, and also made the raw data available as supplementary

information, so the readers can easily access it. The patient have received analgesics in

postoperative period, after appropriate assessment of pain and each given analgesics was

documented. It was not in addition to wound infiltration but if they(bupivacaine alone or a

mixture of bupivacaine and tramadol group) feel pain during postoperative period,

analgesics(rescue analgesics) are given, based on the severity of pain based on patient

management protocol,and finally it was documented and analyzed as total analgesic consumption

in the first 24 hours.

Comment 5: Page 14-the authors write Tramadol may prove predominantly useful in patients

with limited cardiopulmonary function, including the aged, the obese, and smokers, in patients

with impaired liver or renal function, and in patients in whom non-steroidal anti-inflammatory

drugs are not suggested or need to be used with caution (26). However these patient groups are

not included in the current study which focuses on C-Section and according to several sources

tramadol – at least systemic – is not recommended for women who are breast feeding. As I wrote

above - this is an issue that should be discussed at some length if this manuscript is to be

accepted for publicati

---

## [Decision Letter · Decision Letter 1]

24 Jun 2025

Dear Dr. Wonte,

Thank you for submitting your manuscript to PLOS ONE. After careful consideration, we feel that it has merit but does not fully meet PLOS ONE’s publication criteria as it currently stands. Therefore, we invite you to submit a revised version of the manuscript that addresses the points raised during the review process.

=**ACADEMIC EDITOR:**

We look forward to receiving your revised manuscript.

Kind regards,

Regina (Rianne) L.M. van Boekel

Academic Editor

PLOS ONE

Reviewers' comments:

Reviewer's Responses to Questions

**Comments to the Author**

Reviewer #1: All comments have been addressed

Reviewer #2: (No Response)

Reviewer #3: All comments have been addressed

2. Is the manuscript technically sound, and do the data support the conclusions?

Reviewer #1: Yes

Reviewer #2: Partly

Reviewer #3: Yes

3. Has the statistical analysis been performed appropriately and rigorously?

Reviewer #1: Yes

Reviewer #2: I Don't Know

Reviewer #3: Yes

4. Have the authors made all data underlying the findings in their manuscript fully available?

Reviewer #1: Yes

Reviewer #2: No

Reviewer #3: Yes

5. Is the manuscript presented in an intelligible fashion and written in standard English?

Reviewer #1: Yes

Reviewer #2: Yes

Reviewer #3: Yes

Reviewer #1: I am satisfied with the authors' responses and the resulting changes to the paper.........................

Reviewer #2: Mesay Milkias Wonte and colleagues addressed some of the points I raised but not others in my original review. Studies such as this one with vulnerable populations such as women who have given birth and have a responsibility to another person – their baby - have an ethical and medical responsibility to be as medically flawless as possible as other clinicians are liable to emulate their findings. This is one objective of studies. I am referring specifically to what seem to be controversies related to administering tramadol to women who have given birth and that the authors do not address this topic.

Objective of the study: As I wrote in my original review, the authors still do not describe their study as a replication study and validation of infiltrating the CSection incision with a mixture of tramadol and bupivacaine to their setting in Ethiopia. This seems to me a perfectly adequate aim. Though – as I wrote above - am not confident that administering tramadol in this population is medically justified.

Page 10 The authors still do not address the issue of which opioids are recommended vs not for women after C-Section and that there are warnings regarding use of tramadol. The authors assume that when the tramadol is infiltrated into the incision, absorption into the blood is probably not clinically significant. Is this assumption correct in pregnancy when vasculature is more extensive? However, even if this assumption is correct, after surgery, the tramadol was administered systemically. Whether orally or IV is not stated – this should be stated. I, therefore, suggest that the authors cannot avoid addressing the controversies related to tramadol.

Furthermore, it is possible that had they added an opioid to the spinal injection – rather than use a local anaesthesic only and administered other multi-modal treatments, eg 1-2 non-opioid groups round the clock - onset of pain may have been later and pain scores lower. I appreciate that this is a topic for a different study.

Methods

Rather than calculate means - which can be misleading , particularly in studies where pain is assessed - the authors should add the proportion of women who reported scores of ≥3/10 at all the time points assessed. This is missing and the authors compared their findings of pain with the Kumar et al. who assessed the proportion of women who reported scores of >4 at rest and movement-related and did not find differences between the study groups.

The authors did not correct for age – the age range of the sample is large (15 to 49 years) and some underwent at least one CSection. A child of 15 years old who has a CSection will not respond to pain in the same way as a women who is 49 and this is her second or more CSection. I appreciate that the groups are balanced, at least statistically, this respect.

Results

Minor Analgesic consumption – the units – mgs – is missing for average difference of tramadol between the groups. And for diclofenac.

For tramadol - is a mean difference of 10 mg between the groups clinically significant ?

The units of time are also missing under the mean difference for First Analgesia Request Time.

Again – is a mean of 15 minutes difference between the groups, clinically significant?

There is no need to write time at an accuracy of 3 digits after the decimal.

Discussion

Under limitations: that authors should add that assessment of pain after surgery was not multi-modal, it included only assessing pain at rest.

Minor - there is generally no need to repeat the actual findings in terms of data in the Discussion. This is the purpose of the Results section.

Last but one paragraph on page 12 – the authors are probably referring to the systemic doses (IV? Oral?) of the tramadol and diclofenac. Again – no need to include the actual data – this was reported in the Results section.

Reviewer #3: My comments are fully addressed; so, I endorse the acceptance of this paper at this stage as it adds knowledge to the existing literature.

**Do you want your identity to be public for this peer review?** For information about this choice, including consent withdrawal, please see our Privacy Policy

Reviewer #1: No

Reviewer #2: **Yes: ** Ruth Zaslansky, DSc

Reviewer #3: **Yes: ** Dereje Zewdu Assefa

---

## [Author Response · Author response to Decision Letter 2]

3 Jul 2025

Response Letter

Dear Editor,

We would like to thank the editor and reviewers for their careful and thorough reading of this manuscript and for their thoughtful comments and constructive suggestions, which helped us to improve the quality of this manuscript. We have read the research entitled “Analgesic Effectiveness of Wound Infiltration with Bupivacaine versus a Mixture of Bupivacaine and Tramadol for Postoperative Pain Management among Parturients Undergoing Elective Cesarean Section under Spinal Anesthesia”: A Double-Blind Randomized Controlled Trial

We have made all necessary corrections to the comments given by the editor and reviewers one by one as mentioned below. In addition, spelling, grammar, and syntax errors have been corrected.

Editors comments

Comment 1: Not all reviewer comments have been adequately addressed. I recommend that you carefully revisit them.

Response: Thank you, now we have addressed all the comments raised by the reviewers.

Reviewer #1. No comments.

Reviewer #2

Comment 1: Mesay Milkias Wonte and colleagues addressed some of the points, I raised but not others in my original review. Studies such as this one with vulnerable populations such as women who have given birth and have a responsibility to another person – their baby - have an ethical and medical responsibility to be as medically flawless as possible as other clinicians are liable to emulate their findings. This is one objective of studies. I am referring specifically to what seem to be controversies related to administering tramadol to women who have given birth and that the authors do not address this topic.

Response: Thank you for your insightful comments, we totally agree with your concerns, it was also our concern, however a lot of researches conducted in this group of patients shows that the site injection, route of injection, dose, frequency, and degree of metabolism determines the neonatal transfer of the drugs. The excretion of tramadol into milk is low and even lower amounts of the active metabolite, O-desmethyltramadol, are excreted. With usual maternal dosage, the amount excreted into breastmilk is much less than the dose that has been given to newborn infants for analgesia and is unlikely to adversely affect nursing infants(Palmer et al. 2018).

Seventy-five breastfed infants whose mothers were breastfeeding and taking tramadol 100 mg, every 6 hours following a cesarean section were compared to 75 matched infants at 2 to 4 days of age. Forty-nine percent of the mothers taking tramadol and all of the control mothers were taking other opiates (primarily oxycodone) and 61% of and 58%, respectively, also were taking a nonsteroidal anti-inflammatory agent (primarily diclofenac). Examination by a pediatrician revealed no difference between the groups using the Neurologic and Adaptive Capacity Score(K. F. Ilett et al, 2008). In our study tramadol was infiltrated at a wound site which has minimal rate of absorbtion in comparison to intravenous administration, but it significantly reduces pain scores, prolongs pain free state, and on top of that lowers 24-hour analgesia consumption. Here the tramadol was administered in diluted form and infiltrated subcuanseouly which will have minimal plasma concentration in comparison with other routes. Furthermore many studies are conducted in these area specifically on wound infiltration with a combination of opioids with local anesthetics but none of them reported any critical incidents in the mother as well as in the neonate.

Comment 2: Objective of the study: As I wrote in my original review, the authors still do not describe their study as a replication study and validation of infiltrating the C-Section incision with a mixture of tramadol and bupivacaine to their setting in Ethiopia. This seems to me a perfectly adequate aim. Though – as I wrote above - am not confident that administering tramadol in this population is medically justified.

Response: Thank you for your constructive comments, The primary objective of the study was to assesss the severity of pain at the 2nd, 6th, 12th, 18th, and 24th hours. The secondary objectives were total analgesic consumption(mg), the time of the first analgesic request(minutes), incidence of shivering, and post-operative of nausea and vomiting.

Replicating research studies offers several key benefits: validating results, ensuring quality, enhancing generalizability, and advancing scientific knowledge.

Replication helps to confirm that findings are not due to chance or specific conditions and increases confidence in their reliability. It also allows for the identification of potential errors or biases in the original study.

Similar studies were conducted in this population with the same interventions as we have used in developed countries; however no study was conducted in Ethiopia within this specific patient groups, so this randomized controlled trial was mainly aimed to address this issues. On the other hand it helped us to the check the validity and reliability of the previous study.

Validation of this kind of study very important especially in resource limited-areas like our country Ethiopia to decrease the cost related to opioid purchasing frequency for postoperative pain which can have an economic impact on patients and hospitals.

Eventhough tramadol is not recommended in pregnant mothers and lactating mothers, there are multitudes of studies which are conducted in this area including the systematic review but none of them reported any critical fetomaternal incidents associated with the tramadol infiltration at the wound site in normal healthy fit partuirents.

Comment 3: The authors still do not address the issue of which opioids are recommended vs not for women after C-Section and that there are warnings regarding use of tramadol. The authors assume that when the tramadol is infiltrated into the incision, absorption into the blood is probably not clinically significant. Is this assumption correct in pregnancy when vasculature is more extensive? However, even if this assumption is correct, after surgery, the tramadol was administered systemically. Whether orally or IV is not stated – this should be stated. I, therefore, suggest that the authors cannot avoid addressing the controversies related to tramadol.

Furthermore, it is possible that had they added an opioid to the spinal injection – rather than use a local anaesthesic only and administered other multi-modal treatments, eg 1-2 non-opioid groups round the clock - onset of pain may have been later and pain scores lower. I appreciate that this is a topic for a different study.

Response: Thank you for your insightful comments, we totally agree with you regarding the controversies. Generally tramadol is not recommended by FDA for pregnant mothers and lactating mothers, however the neonatal effects of this medication depends on several factors. Infant exposure to drugs in breast milk is dependent on the milk-to-plasma ratio, milk intake, and infant drug clearance (Fríguls B et al, 2010).

By considering this we have selected appropriate parturients in inclusion criteria, term pregnant mothers to make sure that all fetal organs are fully matured, healthy fit mothers without any systemic illinesses to make sure that there is no problem in fetomaternal metabolism of drugs, and no history of opioid usage to make sure that the parturients plasma level of opioid is low or free. Daily infant dose can be calculated as the average concentration of drug in breast milk times the volume of milk ingested in 24 hours or by the milk-to-plasma ratio times maternal plasma concentration times the volume of milk ingested in 24 hours. In the breastfed infant, the estimated volume of milk is 150 mL/kg/day by day 3–4 of life (Salman S, et al. 2011).

We have administered tramadol at wound site which have minimal absorption in comparison to other routes.

The other thing we did to reduce the neonatal drug exposure was exclusion of parturients who were taking opioids in preoperative period to keep the maternal plasma level of opioid as low as possible.

We have stated that the tramadol was given at incision site subcutaneously at a dose of 2mg/kg of tramadol in combination with bupivacaine (0.7mg/kg of 0.25%). None of the patients received tramadol orally or intravenously.

We have searched for evidences regarding if there is any significant differences in subcutaneous changes between lactating mothers and other patient groups interms of vascularity, but no significant variation was observed. According to the studies majority of the changes are anatomical and physiological changes in major organs including, heart, lung, liver, GI, kidney, and brain. The dermatological changes are not significant.

In our study we have not determined the plasma concentration of the drug but we have mentioned it in our limitation section section.

Thank you for your recommendation we will consider addition of opioids to the spinal injections in our future researches.

Comment 4: Rather than calculate means - which can be misleading , particularly in studies where pain is assessed - the authors should add the proportion of women who reported scores of ≥3/10 at all the time points assessed. This is missing and the authors compared their findings of pain with the Kumar et al. who assessed the proportion of women who reported scores of >4 at rest and movement-related and did not find differences between the study groups.

The authors did not correct for age – the age range of the sample is large (15 to 49 years) and some underwent at least one C-Section. A child of 15 years old who has a C-Section will not respond to pain in the same way as a women who is 49 and this is her second or more C-Section. I appreciate that the groups are balanced, at least statistically, this respect.

Response: Thank you for your comments, we totally accept your comments, GEE is generally not ideal for making inferences about individual subjects, as it focuses on population-level effects and this is one of the limitation of GEE.

To address this and avoid misinterpretation we have made the raw data fully accessible as supplementary file, so that the readers can easily access it and can effortlessly visualize and analyze the individual pain score at each time interval. We have analyzed the data with Generalized estimating equations(GEE) which has several advantages including flexibility in correlation structure, handling of missing data robustly, focus on population-Averaged effects, and computational effeciecy. That why we not provided individual pain scores, rather the average pain score at each time interval to exactly determine at which time the parturients feel pain. This average pain score at each time provides us a bias free interpretation of the results. Previously conducted studies analyzed their data with ANOVA which has a limitations when we compare it with the Generalized estimating equations.

We have selected this age group (15-49) because the WHO categorized this age group as fertile age group. The parturients were allocated randomly to each group with computer generated codes and the data collectors were blinded to avoid selection bias. The socio-demographic characteristics of the patients including age was calculated but it was not statistically significant (P=0.45). That is why have not used correct for age.

Regarding the number previous cesarean section, we totally agree with you and included it in the limitation section.

Comment 5: Minor Analgesic consumption – the units – mgs – is missing for average difference of tramadol between the groups. And for diclofenac. For tramadol - is a mean difference of 10 mg between the groups clinically significant ? The units of time are also missing under the mean difference for First Analgesia Request Time. Again – is a mean of 15 minutes difference between the groups, clinically significant? There is no need to write time at an accuracy of 3 digits after the decimal.

Response: Thank you for your detailed comment, we have corrected the units and milligrams.

We have calculated effect size to determine the clinical significance(d=0.84), which is large effect size. To say clinically significant we have calculated the effect size(95% CI, 13.45 to 56.54), t (58) = 3.252, P= 0.002, d=0.84), not the mean difference. That is why we said it is clinically significant.

Sorry, here we said it is statistically significant (95% CI, -182.087 to -119.913), t (58) = 5.6553, P= 0.001) not clinically significant. Because we have considered that p-value is statistically significant when the p-value less than 0.05.

Thanks in advance we have corrected the decimals.

Comment 6: Under limitations: that authors should add that assessment of pain after surgery was not multi-modal, it included only assessing pain at rest. Minor - there is generally no need to repeat the actual findings in terms of data in the Discussion. This is the purpose of the Results section. Last but one paragraph on page 12 – the authors are probably referring to the systemic doses (IV? Oral?) of the tramadol and diclofenac. Again – no need to include the actual data –this was reported in the Results section.

Response: Thank you for your comments, we have added it in limitation section. Thanks in advance for your comments majority of journals including this journal recommends to describe the main findings of primary outcome variables in discussion section, that is why we have added some findings of primary outcome variables of research in the discussion section.

In page 12, “This finding might be due to tramadol with bupivacaine acting synergistically, which might have provided lower analgesic consumption.” This statement was to describe that during wound infiltration with a mixture of tramadol and bupivacaine may act synergistically, that synergistic effect may prolong the analgesic duration, and believed to result in pain free period. It was not intravenous or oral route.

The actual data was included, because according to the PLOS ONE submission guideline each and every underlying data have to be accessible freely for the public. That why we have included the actual data.

Reviewer #3: No comments.

Regards

Mesay Milkias,

Semagn Mekonnen,

Hailemariam Getachew,

Hailemariam Mulugeta,

Siraj Ahmed,

Melkamu Kebede,

Belete Destaw,

Medhanit Melese,

Zemedu Aweke

---

## [Decision Letter · Decision Letter 2]

28 Jul 2025

Dear Dr. Wonte,

Thank you for submitting your manuscript to PLOS ONE. After careful consideration, we feel that it has merit but does not fully meet PLOS ONE’s publication criteria as it currently stands. Therefore, we invite you to submit a revised version of the manuscript that addresses the points raised during the review process.

We look forward to receiving your revised manuscript.

Kind regards,

Dereje Zewdu Assefa, BSc, MSc

Academic Editor

PLOS ONE

Journal Requirements:

Additional Editor Comments:

The manuscript has improved during the second round of revisions. However, the second reviewers raised critical questions that the authors need to address.

Reviewers' comments:

Reviewer's Responses to Questions

**Comments to the Author**

Reviewer #1: All comments have been addressed

Reviewer #2: (No Response)

Reviewer #3: All comments have been addressed

2. Is the manuscript technically sound, and do the data support the conclusions?

Reviewer #1: Yes

Reviewer #2: Partly

Reviewer #3: Yes

3. Has the statistical analysis been performed appropriately and rigorously?

Reviewer #1: Yes

Reviewer #2: N/A

Reviewer #3: Yes

4. Have the authors made all data underlying the findings in their manuscript fully available?

Reviewer #1: Yes

Reviewer #2: (No Response)

Reviewer #3: Yes

5. Is the manuscript presented in an intelligible fashion and written in standard English?

Reviewer #1: Yes

Reviewer #2: Yes

Reviewer #3: Yes

Reviewer #1: No further comments..........................................................................................

Reviewer #2: The authors continue to avoid addressing the controversies related to tramadol in women who are pregnant and plan to breastfeed. The studies cited by the authors as indicating that tramadol is safe were not powered to detect side effects. The authors should be transparent about the use of tramadol in this cohort of women. Particularly if they wish that this form of treatment is to be employed by other clinicians.

In their reply the authors write:

>Even though tramadol is not recommended in pregnant mothers and lactating mothers, there are multitudes of studies >which are conducted in this area including the systematic review but none of >them reported any critical fetomaternal >incidents associated with the tramadol infiltration at the >wound site in normal healthy fit partuirents.

The authors admit that tramadol is not recommended in pregnant women and lactating mothers.

The study has now been carried out and if the editors of this journal wish to publish it – it does not seem ethical to me to avoid addressing the complexities of administering tramadol in this cohort of women even when the medication is infiltrated to the surgical incision. Availability of other published studies – not powered to detect side effects – is not justification to continue along these lines.

Furthermore, it is possible that women received tramadol systemically after the surgery. This is still unclear to me. Needs to be clarified. See below.

The authors write that

>None of the patients received tramadol orally or intravenously.

How was the tramadol (and diclofenac) after surgery administered? I cannot find that this is addressed in the Methods but it is in the Results, the authors write.

>Analgesia consumption

>The mean tramadol consumption was lower in a mixture of tramadol and bupivacaine group (BT) (140.00+48.06 mg) >than bupivacaine group (175.00+34.11 mg) with a statistically significant mean difference(MD) of 10.76 (95% CI, 13.45 >to 56.54), t (58) = 3.25, P= 0.002, (d=0.84). There was no significant difference in mean diclofenac consumption >between the two groups in a combination of tramadol and bupivacaine group (52.50+48.84 mg) than bupivacaine alone >group (57.50+46.95 mg), (P=0.69), (d=0.10).

How was this tramadol administered ?

In the discussion the authors write

>The efficacy of tramadol for the management of moderate to severe postoperative pain has been revealed in both >inpatients and day surgery patients. Most importantly, contrasting with other opioids, tramadol has no clinically >significant effects on respiratory or cardiovascular parameters. Tramadol may prove predominantly useful in patients >with limited cardiopulmonary function, including the aged, the obese, and smokers; in patients with impaired liver or >renal function; and in >patients in whom non-steroidal anti-inflammatory drugs are not suggested or need to be used with >caution (37).

This paragraph bears little relevance to women who have given birth.

The proportion of patients who report scores of 4/10 and above

I suggest that the authors create a table with the % of women in both groups whose scores are 4/10, displaying this information over time. This is a simple analysis to carry out and should not be expected of readers.

To compare the pain scores, they might use Mann-Whitney U test and correct for multiple comparisons.

The table can be included in the appendix as this is not the primary analysis.

In the Discussion, the authors cite the study carried out by Kumar M, Batra YK, Panda NB, Rajeev S, Nagi ON. Tramadol added to bupivacaine does not prolong analgesia of continuous psoas compartment block . Pain Pract. 2009;9(1):43-50.

These were patients undergoing total hip replacement and received a different block - and so I am not sure how this citation is relevant for discussion of the findings in the current study, unless the authors intend to review different procedures where bupivacaine and tramadol are used for infusion. Please clarify.

Minor.

Replication study. The authors address the strengths of a replication study in their reply to the editor but do not address this in their manuscript. Women undergoing CSection in many countries in the world, would benefit from improved treatment and so it is laudable that the authors are seeking better care for women in Ethiopia.

Reviewer #3: My comments are fully addressed. Given that this study adds knowledge to the existing literature, especially in resource-limited settings, I fully endorse the acceptance of this manuscript.

**Do you want your identity to be public for this peer review?** For information about this choice, including consent withdrawal, please see our Privacy Policy

Reviewer #1: No

Reviewer #2: **Yes: ** Do you want your identity to be public for this peer review? My identity can be public, for the sake of transparency. I am not seeking the recognition.

Reviewer #3: **Yes: ** Dereje Zewdu Assefa

---

## [Author Response · Author response to Decision Letter 3]

18 Aug 2025

Response Letter

Dear Editor,

We would like to thank the editor and reviewers for their careful and thorough reading of this manuscript and for their thoughtful comments and constructive suggestions, which helped us to improve the quality of this manuscript. We have read the research entitled “Analgesic Effectiveness of Wound Infiltration with Bupivacaine versus a Mixture of Bupivacaine and Tramadol for Postoperative Pain Management among Parturients Undergoing Elective Cesarean Section under Spinal Anesthesia”: A Double-Blind Randomized Controlled Trial

We have made all necessary corrections to the comments given by the editor and reviewers one by one as mentioned below. In addition, spelling, grammar, and syntax errors have been corrected.

Editors comments

Comment 1: The manuscript has improved during the second round of revisions. However, the second reviewers raised critical questions that the authors need to address.

Response: Thank you, now we have addressed all the comments raised by the second reviewer.

Reviewer #1. No comments.

Reviewer #2

Comment 1: The authors continue to avoid addressing the controversies related to tramadol in women who are pregnant and plan to breastfeed. The studies cited by the authors as indicating that tramadol is safe were not powered to detect side effects. The authors should be transparent about the use of tramadol in this cohort of women. Particularly if they wish that this form of treatment is to be employed by other clinicians.

Response: Thank you for your insightful comments, We appreciate the reviewer’s concern regarding tramadol use in lactating women. As noted, tramadol is not routinely recommended during breastfeeding due to potential neonatal risks. However, its use in our study was limited to a short duration in a setting where alternative opioids were unavailable. All mothers and neonates were closely monitored for difficulty in breast feeding, breathing difficulties, and drowsiness in the baby and no adverse effects were observed. We have added clarification in the manuscript to reflect this context and referenced relevant safety guidelines. This clinical trial may have significant contribution especially in resources limited set-ups, by minimizing the economic burden and side effects associated with higher dosages of opioids.

Comment 2: The authors admit that tramadol is not recommended in pregnant women and lactating mothers. The study has now been carried out and if the editors of this journal wish to publish it – it does not seem ethical to me to avoid addressing the complexities of administering tramadol in this cohort of women even when the medication is infiltrated to the surgical incision. Availability of other published studies – not powered to detect side effects – is not justification to continue along these lines. Furthermore, it is possible that women received tramadol systemically after the surgery. This is still unclear to me. Needs to be clarified.

Response: Thank you for your constructive comments, tramadol is not routinely recommended during breastfeeding due to potential neonatal risks, however this neonatal risks are minimized by minimizing the duration of drug exposure, reducing the dosage or administering lowest effective dose, and choosing the appropriate routes which have least absorption to breast milk are recommended. Almost all drugs are transferred into the breast milk to some extent. Infant exposure to drugs in breast milk is dependent on the milk-to-plasma ratio, milk intake, and infant drug clearance(Fríguls B., Joya X., García-Algar O., Pallás C.R., Vall O., Pichini S. (2010) A comprehensive review of assay methods to determine drugs in breast milk and the safety of breastfeeding when taking drugs. Analytical and Bioanalytical Chemistry, 397, 1157–1179).

A trial in women who received a cesarean section randomized participants to receive 2 tablets of acetaminophen 325 mg and tramadol 35.5 mg (Zaldiar, Gruenthal, Israel) every 6 hours as well as diclofenac 100 mg at 12, 24 and 48 hours after arrival in the maternity ward, either as a fixed dose or the same drugs on demand. No differences were noted in neonatal outcomes of peak bilirubin, need for phototherapy, and adverse effects, including irritability or altered consciousness. No adverse infant effects were seen(Yefet E, Taha H, Salim R, et al. Fixed time interval compared with on-demand oral analgesia protocols for post-caesarean pain: A randomised controlled trial. BJOG 2017;124:1063–70.).

A hospital in Japan provided postpartum mothers with a combination tablet containing tramadol 37.5 mg and acetaminophen 325 mg every 6 hours for 3 days if they requested a pain medication. A retrospective analysis found that of 148 mothers who received the drug, all infants were breastfed and none of the infants had an adverse reaction, such as drowsiness, difficulty breastfeeding or breathing problems(Nagae Seika P, Taizan K, Mariko S, et al. Safety of tramadol hydrochloride/acetaminophen combination tablet use for postpartum pain. J Obstet Gynaecol Res 2021;47:2919.). The relatively low dose and short duration of therapy may have reduced the risk of adverse reactions.

Others clinical trial reported infants were exposed to <3% of a mother’s tramadol dose through breast milk with no evidence of harmful effects and suggested short-term maternal use was compatible with breastfeeding, This clinical study reported a milk-to-plasma ratio of 2.2 and 2.8 for tramadol and O-desmethyltramadol, respectively. Additionally, they reported a relative infant dose of 2.24 and 0.64% for tramadol and O-desmethyltramadol, respectively(Ilett K.F., Paech M.J., Page-Sharp M., Sy S.K., Kristensen J.H., Goy R., et al. (2008) Use of a sparse sampling study design to assess transfer of tramadol and its O-desmethyl metabolite into transitional breast milk. British Journal of Clinical Pharmacology, 65, 661–666.).

Comment 3: How was the tramadol (and diclofenac) after surgery administered? I cannot find that this is addressed in the Methods but it is in the Results, the authors write. How was this tramadol administered ?

Response: Thank you for your comment, In our study tramadol was used in the following scenarios:-

1. Following the surgery, the wound site was in the lower abdomen and transverse and about 1-2 inches above the pubic hairline, was infiltrated with 0.7 mg/kg of 0.25% bupivacaine diluted to 20 ml for group B. The BT group received a mixture of 0.7 mg/kg of 0.25% bupivacaine and 2 mg/kg of tramadol diluted to 20 milliliters by the four experienced obstetricians.

2. After the administration spinal anesthesia the grading of post-spinal shivering was assessed with Bedside Shivering Assessment Score (BSAS) and those patients whom develop shivering with BSAS grade greater or equal to 3 received intravenous route (IV) tramadol 0.5 mg/kg. The tramadol which adminstred for shivering may have little effect on our intervention, so we have mentioned it in the limitation section.

3. The mean tramadol consumption was lower in a mixture of tramadol and bupivacaine group (BT) (140.00+48.06 mg) than bupivacaine group (175.00+34.11 mg) with a statistically significant mean difference(MD) of 10.76 (95% CI, 13.45 to 56.54), t (58) = 3.25, P= 0.002, (d=0.84). Here tramadol was intravenously. There was no significant difference in mean diclofenac consumption between the two groups in a combination of tramadol and bupivacaine group (52.50+48.84 mg) than bupivacaine alone group (57.50+46.95 mg), (P=0.69), (d=0.10). Diclofenac was given intramuscular route(IM). Here we have mentioned the total analgesic given, because one of our secondary outcome was the total analgesia consumption.

Comment 4: The proportion of patients who report scores of 4/10 and above

I suggest that the authors create a table with the % of women in both groups whose scores are 4/10, displaying this information over time. This is a simple analysis to carry out and should not be expected of readers. To compare the pain scores, they might use Mann-Whitney U test and correct for multiple comparisons. The table can be included in the appendix as this is not the primary analysis.

Response: Thank you for your comments, we have added the table as supplementary file and corrected it; however we have used NRS to assess the severity of pain in our study which is categorical(Mild(1-3), Moderate(4-6), and Severe(7-10)), that is why have not provided the exact score as you mentioned above (4/10). In our study those parturients having pain score 4/10 is categorized under moderate.

Comment 5: In the Discussion, the authors cite the study carried out by Kumar M, Batra YK, Panda NB, Rajeev S, Nagi ON. Tramadol added to bupivacaine does not prolong analgesia of continuous psoas compartment block . Pain Pract. 2009;9(1):43-50. These were patients undergoing total hip replacement and received a different block - and so I am not sure how this citation is relevant for discussion of the findings in the current study, unless the authors intend to review different procedures where bupivacaine and tramadol are used for infusion. Please clarify.

Response: Thank you for your comment, we have removed it.

Comment 6: Replication study. The authors address the strengths of a replication study in their reply to the editor but do not address this in their manuscript. Women undergoing CSection in many countries in the world, would benefit from improved treatment and so it is laudable that the authors are seeking better care for women in Ethiopia.

Response: Thank you for your comments, we have added it.

Reviewer #3: No comments.

Regards

Mesay Milkias,

Semagn Mekonnen,

Hailemariam Getachew,

Hailemariam Mulugeta,

Siraj Ahmed,

Melkamu Kebede,

Belete Destaw,

Medhanit Melese,

Zemedu Aweke

---

## [Decision Letter · Decision Letter 3]

10 Sep 2025

Dear Dr. Wonte,

Thank you for submitting your manuscript to PLOS ONE. After careful consideration, we feel that it has merit but does not fully meet PLOS ONE’s publication criteria as it currently stands. Therefore, we invite you to submit a revised version of the manuscript that addresses the points raised during the review process.

We look forward to receiving your revised manuscript.

Kind regards,

Dereje Zewdu Assefa, BSc, MSc

Academic Editor

PLOS ONE

Journal Requirements:

Additional Editor Comments :

The manuscript has notably improved following the third round of revisions, and the authors have addressed several previous concerns with greater clarity. However, Reviewer 2 has raised important and critical points that still require your attention prior to acceptance.

In particular, the safety of tramadol use in breastfeeding mothers remains insufficiently addressed. Please provide a clear, transparent, and evidence-based discussion on the potential risks associated with tramadol exposure in neonates through breast milk. This should include documented effects such as respiratory depression or central nervous system depression, supported by relevant and current literature. Ensure that this safety consideration is prominently discussed in both the manuscript and any clinical recommendations derived from your findings.

Additionally, please upload the supplementary table as referenced in your response letter or manuscript, as it is currently missing from the submission.

We look forward to receiving a revised version that adequately addresses these final points.

Reviewers' comments:

Reviewer's Responses to Questions

**Comments to the Author**

Reviewer #2: (No Response)

2. Is the manuscript technically sound, and do the data support the conclusions?

Reviewer #2: No

3. Has the statistical analysis been performed appropriately and rigorously?

Reviewer #2: N/A

4. Have the authors made all data underlying the findings in their manuscript fully available?

Reviewer #2: Yes

5. Is the manuscript presented in an intelligible fashion and written in standard English?

Reviewer #2: Yes

Reviewer #2: Reply to the third revision

There is an ongoing to and fro with the authors about the safely of administering tramadol for lactating women.

As studies suggest that a significant proportion of people in Ethiopia may be CYP2D6 ultra-rapid metabolizers, medications such as tramadol and codeine increase the risk of adverse effects, especially in breastfeeding infants.

The authors write that they ‘We have added clarification in the manuscript to reflect this context and

referenced relevant safety guidelines’

Where are these citations? .

Recent guidelines such as the 2021 NICE guidelines: Caesarean birth [F] Opioids for pain relief after caesarean birth

https://www.nice.org.uk/guidance/ng192/evidence/f-opioids-for-pain-relief-after-caesarean-birth-pdf-9071941651

Opioid analgesia for women who have had a spinal/epidural anaesthesia: The committee noted that (in studies which used spinal anaesthesia) **** morphine was more effective than pethidine (also known as meperidine) for pain relief and had less impact on breastfeeding.**** Oral oxycodone was more effective than IV morphine or IV oxycodone at reducing the incidence of moderate and severe pain, with less nausea and vomiting. However, the committee discussed that the FDA and American Academy of Paediatrics advise that oxycodone (as well as codeine and tramadol) increases the risk of neonatal sedation and respiratory depression, and that oral morphine or the less commonly-used 18 Caesarean birth: evidence reviews for opioids as pain relief FINAL (March 2021) FINAL Opioids for pain relief hydromorphone may be suitable alternatives. In addition, the MHRA has issued a warning advising that codeine should not be taken by breastfeeding mothers. ****The committee noted that codeine and tramadol can be particularly problematic in up to 28% of women who are CYP2D6 ultra-rapid metabolisers and who convert these drugs to morphine rapidly, leading to high morphine levels in their breast milk.****

The FDA warn against use of tramadol

Codeine and Tramadol Medicines: Drug Safety Communication - Restricting Use in Children, Recommending Against Use in Breastfeeding Women | American Society of Anesthesiologists (ASA) April 27, 2017

• A strengthened Warning to mothers that breastfeeding is not recommended when taking codeine or tramadol medicines due to the risk of serious adverse reactions in breastfed infants. These can include excess sleepiness, difficulty breastfeeding, or serious breathing problems that could result in death.

Text such as the authors write in their reply to the reviewers should be included in the article itself. For example: Tramadol is not routinely recommended during breastfeeding due to potential neonatal risks (REFS), this is particularly relevant in CYP2D6 ultra-rapid metabolizers, however, we (the authors) suggest that these neonatal risks are minimized by minimizing the duration of drug exposure, reducing the dosage or administering lowest effective dose, and choosing the appropriate routes which have least absorption to breast milk are recommended’.

Details from the FDA Communication (April 20, 2017)

• Strengthened Warning: The FDA added a Contraindication to tramadol’s label, explicitly advising against its use during breastfeeding for safety reasons U.S. Food and Drug AdministrationASA.

• Explanation: Tramadol and its active metabolite (O desmethyltramadol) are present in breast milk. Mothers who are CYP2D6 ultra rapid metabolizers may convert tramadol more quickly to its active form, increasing the risk of infant overdose via breast milk U.S. Food and Drug Administration+1.

• FDA Recommendation: The agency urges both healthcare professionals and breastfeeding women to avoid codeine and tramadol and to consider safer alternative

How was the tramadol (and diclofenac) after surgery administered? I

cannot find that this is addressed in the Methods but it is in the Results, the authors

write. How was this tramadol administered.

According to the authors:

1. It was infiltrated into the wound

2. For shivering – not clear was tramadol or pethidine administered for shivering ?

3. After surgery – IV tramadol with IM diclofenac. Why is this not stated clearly in the methods?

The proportion of patients who report scores of 4/10 and above.

I have not been able to access the supplementary table.

**Do you want your identity to be public for this peer review?** For information about this choice, including consent withdrawal, please see our Privacy Policy

Reviewer #2: **Yes: ** Ruth Zaslansky, DSc

---

## [Author Response · Author response to Decision Letter 4]

22 Sep 2025

Response Letter

Dear Editor,

We would like to thank the editor and reviewers for their careful and thorough reading of this manuscript and for their thoughtful comments and constructive suggestions, which helped us to improve the quality of this manuscript. We have read the research entitled “Analgesic Effectiveness of Wound Infiltration with Bupivacaine versus a Mixture of Bupivacaine and Tramadol for Postoperative Pain Management among Parturients Undergoing Elective Cesarean Section under Spinal Anesthesia”:A Double-Blind Randomized Controlled Trial

We have made all necessary corrections to the comments given by the editor and reviewers one by one as mentioned below. In addition, spelling, grammar, and syntax errors have been corrected.

Editors comments

Comment 1: The manuscript has notably improved following the third round of revisions, and the authors have addressed several previous concerns with greater clarity. However, Reviewer 2 has raised important and critical points that still require your attention prior to acceptance.

In particular, the safety of tramadol use in breastfeeding mothers remains insufficiently addressed. Please provide a clear, transparent, and evidence-based discussion on the potential risks associated with tramadol exposure in neonates through breast milk. This should include documented effects such as respiratory depression or central nervous system depression, supported by relevant and current literature. Ensure that this safety consideration is prominently discussed in both the manuscript and any clinical recommendations derived from your findings.

Additionally, please upload the supplementary table as referenced in your response letter or manuscript, as it is currently missing from the submission.

Response: Thank you; now we have addressed all the comments raised by the second reviewer regarding the safety of tramadol. Additionally, we have added a supplementary table that was missed during submission unintentionally. We greatly apologize for that.

Reviewer #1. No comments.

Reviewer #2

Comment 1: There is an ongoing to and fro with the authors about the safely of administering tramadol for lactating women.

As studies suggest that a significant proportion of people in Ethiopia may be CYP2D6 ultra-rapid metabolizers, medications such as tramadol and codeine increase the risk of adverse effects, especially in breastfeeding infants.

Response: Thank you for your insightful comments. We appreciate the reviewer’s concern regarding tramadol use in lactating women. As noted, tramadol is not routinely recommended during breastfeeding due to potential neonatal risks. However, its use in our study was limited to a short duration in a setting where alternative opioids were unavailable. All mothers and neonates were closely monitored for difficulty in breastfeeding, breathing difficulties, and drowsiness in the baby, and no adverse effects were observed. We have added clarification in the manuscript to reflect this context and referenced relevant safety guidelines as follows:

The safety of tramadol in neonates is limited due to the small size of the available data. Plasma concentrations of tramadol and its metabolite M1 in neonates, after breastfeeding, are influenced by three main factors: the amount of tramadol and M1 present in breast milk (which depends on maternal drug concentration, maternal metabolism, diffusion, ion trapping, and lipid partitioning), the quantity of breast milk consumed and its bioavailability, and the neonatal clearance of these compounds (Palmer GM, Anderson BJ, Linscott DK, Paech MJ, Allegaert K. Tramadol, breastfeeding, and safety in the newborn. Archives of disease in childhood. 2018;103(12):1110-3.).

It is known that the amount of tramadol excreted into early breast milk is a small percentage of the maternal dose, specifically less than 2.5% per kg of body weight per day. In a study, seventy-five breastfeeding mothers were administered tramadol at a dose of 100 mg every 6 hours following a cesarean birth, in this study exposed infants and the control group of breastfed infants had similar characteristics, such as Apgar scores at birth and scores for neurological and adaptive capacities (Ilett KF, Paech, M. J., Page-Sharp, M., Sy, S. K., Kristensen, J. H., Goy, R., Chua, S., Christmas, T., & Scott, K. L. (2008). Use of a sparse sampling study design to assess transfer of tramadol and its O-desmethyl metabolite into transitional breast milk. British journal of clinical pharmacology,. 2008;65(5):661–6.).

The neonate’s plasma concentration was 35 times higher than the upper limit found in breastfed infants of mothers prescribed 60 mg of codeine for postpartum pain (0.5–2.2 mcg/L) and 3.5 times the threshold that can cause respiratory depression in neonates (20 mcg/L). A review of the literature on tramadol use during breastfeeding has not identified any similar adverse events ((MD): DaLDLIB. National Institute of Child Health and Human Development; 2006- Tramadol [Updated 2025 Aug 15]. 2006.). While both tramadol and its M1 metabolite are present in breast milk, there is no evidence suggesting they carry the same risks associated with the ultra-rapid metabolism of codeine (Palmer GM, Anderson BJ, Linscott DK, Paech MJ, Allegaert K. Tramadol, breastfeeding, and safety in the newborn. Archives of disease in childhood. 2018;103(12):1110-3.).

Several studies have been conducted regarding the safety of opioids in lactating mothers; however, they have small sample sizes, wide confidence intervals, and a high risk of bias (https://www.nice.org.uk/guidance/ng192/evidence/f-opioids-for-pain-relief-after caesarean-birth-pdf-9071941651)

In addition to this we have added DNA testing at limitation section to determine those patients which are ultrametabolizers in future studies.

This clinical trial may have significant contribution to patients and family especially in resources limited set-ups like our country, by minimizing the economic burden and side effects associated with higher dosages of opioids.

Comment 3: How was the tramadol (and diclofenac) after surgery administered? I cannot find that this is addressed in the Methods but it is in the Results, the authors write. How was this tramadol administered. According to the authors:

1. It was infiltrated into the wound

2. For shivering – not clear was tramadol or pethidine administered for shivering ?

3. After surgery – IV tramadol with IM diclofenac. Why is this not stated clearly in the methods?

Response: Thank you for your comment, we have clearly stated how and when tramadol and diclofenac were used in our study. In our study tramadol was used in the following scenarios:

1. Following the surgery, the wound site, which was in the lower abdomen and transverse and about 1-2 inches above the pubic hairline, was infiltrated with 0.7 mg/kg of 0.25% bupivacaine diluted to 20 ml for group B. The BT group received a mixture of 0.7 mg/kg of 0.25% bupivacaine and 2 mg/kg of tramadol diluted to 20 milliliters by the four experienced obstetricians. In this section tramadol was added to bupivacaine for the intervention group. (Page 6)

2. After the administration spinal anesthesia the grading of post-spinal shivering was assessed with the Bedside Shivering Assessment Score (BSAS), and those patients who developed shivering with a BSAS grade greater than or equal to 3 received intravenous (IV) tramadol 0.5 mg/kg. The tramadol administered for shivering may have little effect on our intervention, so we have mentioned it in the limitation section. Tramadol was used for shivering. In this section tramadol was given for shivering. (Page 6)

3. Thank you for the observation; it was an editorial problem. We greatly apologize for that, and we have corrected it. Standard care was provided equally for both groups postoperatively. Rescue analgesics were given based on the severity of pain, and the total analgesics (tramadol and diclofenac) administered during the first 24 hours are documented and analyzed as a secondary outcome.(Page 6)

Reviewer #3: No comments.

Regards

Mesay Milkias,

Semagn Mekonnen,

Hailemariam Getachew,

Hailemariam Mulugeta,

Siraj Ahmed,

Melkamu Kebede,

Belete Destaw,

Medhanit Melese,

Zemedu Aweke

---

## [Decision Letter · Decision Letter 4]

26 Oct 2025

Analgesic Effectiveness of Wound Infiltration with Bupivacaine versus a Mixture of Bupivacaine and Tramadol for Postoperative Pain Management among Parturients Undergoing Elective Cesarean Section under Spinal Anesthesia:A Randomized Controlled Trial

PONE-D-25-04955R4

Dear Dr. Wonte,

We’re pleased to inform you that your manuscript has been judged scientifically suitable for publication and will be formally accepted for publication once it meets all outstanding technical requirements.

Kind regards,

Dereje Zewdu Assefa, BSc, MSc

Academic Editor

PLOS ONE

Additional Editor Comments (optional):

Reviewers' comments:

Reviewer's Responses to Questions

**Comments to the Author**

Reviewer #1: All comments have been addressed

Reviewer #2: All comments have been addressed

2. Is the manuscript technically sound, and do the data support the conclusions?

Reviewer #1: Yes

Reviewer #2: Partly

3. Has the statistical analysis been performed appropriately and rigorously?

Reviewer #1: Yes

Reviewer #2: No

4. Have the authors made all data underlying the findings in their manuscript fully available?

Reviewer #1: Yes

Reviewer #2: Yes

5. Is the manuscript presented in an intelligible fashion and written in standard English?

Reviewer #1: Yes

Reviewer #2: Yes

Reviewer #1: Again, there is nothing else for me to add. Again, there is nothing else for me to add. Again, there is nothing else for me to add.

Reviewer #2: This is the 4th round of revisions.

The editors have decided to accept this manuscript.

I have no further comments.

**Do you want your identity to be public for this peer review?** For information about this choice, including consent withdrawal, please see our Privacy Policy

Reviewer #1: No

Reviewer #2: **Yes: ** Ruth Zaslansky, DSc

---

## [Editor Report · Acceptance letter]

PONE-D-25-04955R4

PLOS ONE

Dear Dr. Wonte,

I'm pleased to inform you that your manuscript has been deemed suitable for publication in PLOS ONE. Congratulations! Your manuscript is now being handed over to our production team.

Kind regards,

on behalf of

Professor Dereje Zewdu Assefa

Academic Editor

PLOS ONE